# Local community assembly mechanisms shape soil bacterial β diversity patterns along a latitudinal gradient

Xiao Zhang [1,2], Shirong Liu [1✉], Jingxin Wang[3], Yongtao Huang [2], Zachary Freedman[4], Shenglei Fu[2], Kuan Liu [5], Hui Wang[1], Xiangzhen Li[6], Minjie Yao[6], Xiaojing Liu[7] & Jamie Schuler[3]

Biodiversity patterns across geographical gradients could result from regional species pool and local community assembly mechanisms. However, little has been done to separate the effects of local ecological mechanisms from variation in the regional species pools on bacterial diversity patterns. In this study, we compare assembly mechanisms of soil bacterial communities in 660 plots from 11 regions along a latitudinal gradient in eastern China with highly divergent species pools. Our results show that β diversity does not co-vary with γ diversity, and local community assembly mechanisms appear to explain variation in β diversity patterns after correcting for variation in regional species pools. The variation in environmental conditions along the latitudinal gradient accounts for the variation in β diversity through mediating the strength of heterogeneous selection. In conclusion, our study clearly illustrates the importance of local community assembly processes in shaping geographical patterns of soil bacterial β diversity.

---

[1] Key Laboratory of Forest Ecology and Environment, China's State Forestry and Grassland Administration, Institute of Forest Ecology, Environment and Protection, Chinese Academy of Forestry, Beijing, China. [2] Key Laboratory of Geospatial Technology for the Middle and Lower Yellow River Regions, Ministry of Education. College of Environment and Planning, Henan University, Jinming Avenue, Kaifeng, China. [3] Division of Forestry and Natural Resources, West Virginia University, Morgantown, WV 26506-6125, USA. [4] Division of Plant and Soil Sciences, West Virginia University, Morgantown, WV 26506-6125, USA. [5] Dalla Lana School of Public Health, University of Toronto, 155 College Street, Toronto, ON M5T 3M7, Canada. [6] Fujian Provincial Key Laboratory of Soil Environmental Health and Regulation, College of Resources and Environment, Fujian Agriculture and Forestry University, Fuzhou 350002, China. [7] Baotianman Natural Reserve Administration, Neixiang 474350, China. ✉email: liusr@caf.ac.cn

Molecular phylogenetic analysis of the geographic distributions of microorganisms indicates that microbial taxa exhibit different biogeographic patterns relative to macro-organisms[1–4]. This highlights unique features of microbial life that determine the generation and maintenance of biodiversity[1,5]. Disentangling the mechanisms underlying biogeographic patterns is the key to ascertain why these differences exist. For example, latitude may not serve equally as well as a proxy for macrobial and microbial diversity[3,6–8]. One possible explanation for this pattern is that assembly processes at large scales (i.e., continental and global) that create differences in regional species pools may drive biogeographical gradients[9–11]. Alternatively, variation in diversity across large scale geographical gradients could result from the differences in community assembly processes at local scale[12]. Most previous studies focused on the influences of large scale assembly processes on biogeographical gradients in β diversity[1,4,13], yet few studies explored β-diversity patterns underpinned by local community assembly processes and regional species pools along latitudinal gradients.

Patterns of site-to-site variation in species composition, known as β diversity, not only describes the scaling relationship between local (α) and regional (γ) diversity, but also provides fundamental insights into the processes that create and maintain biodiversity[14,15]. This commonly studied pattern has been used to illuminate biogeographic distributions of both microorganisms and macroorganisms along environmental gradients[10,13]. Some studies indicate that variation in β diversity is mediated primarily by variation in regional species pool rather than community assembly processes[10,16]. However, other studies have shown that variation in β diversity is more likely to be driven by community assembly mechanisms (e.g., niche versus neutral)[8,12,13,15]. For example, a heterogeneous environment can influence community composition by selectively filtering species from the regional species pool in different ways across local communities, resulting in high β diversity[17]. Here, changes of environmental factors, such as, soil pH[18] and carbon[7], can promote turnover in soil

bacterial community composition[17]. Alternatively, a homogeneous environment can result in low β diversity by selectively filtering species in a similar way[16]. Moreover, variation in β diversity may also depend on dispersal processes[15,16]. High dispersal rates such as homogenizing dispersal[12] may decrease β diversity and overall species diversity by increasing the distributions of dominant species[19,20], while dispersal limitation would result in aggregated distributions of species in space[21], such that closer sites will have more similar community composition than more geographically separated sites, which would increase β diversity across locations. In addition, stochastic processes, species interactions or priority effects may also influence β diversity by changing the number of individuals or species in a community[1,22].

Recent studies on soil bacterial β diversity patterns show that the similarity of bacterial communities declines with increasing geographic distance and environmental heterogeneity[8,13,23], and that variation in bacterial β diversity is strongly correlated with variation in environmental factors including soil pH, soil C and N contents[13,17,18]. As environmental variables often vary along a latitudinal gradient[7], this could be the main cause of variation in local community assembly. Also, environmental variables influence the relative importance of deterministic and stochastic processes across different regions. Alternatively, variation in the regional species pool itself could also result in variation in β diversity wherein γ diversity and species distribution significantly vary among different regions[10,11]. Thus, there is a clear need to distinguish the influence of local community assembly processes from regional species pool, and this will help to ascertain the mechanism underlying the geographical pattern of soil bacterial community.

In this study, we conduct a 3700 km long transect survey, including 11 typical regions along a north-south transect of Eastern China, which covers a wide range of latitudes, a great variety of climate types (from tropical to boreal zones, Fig. 1 and Supplementary Fig. 1) as well as a full spectrum of environmental

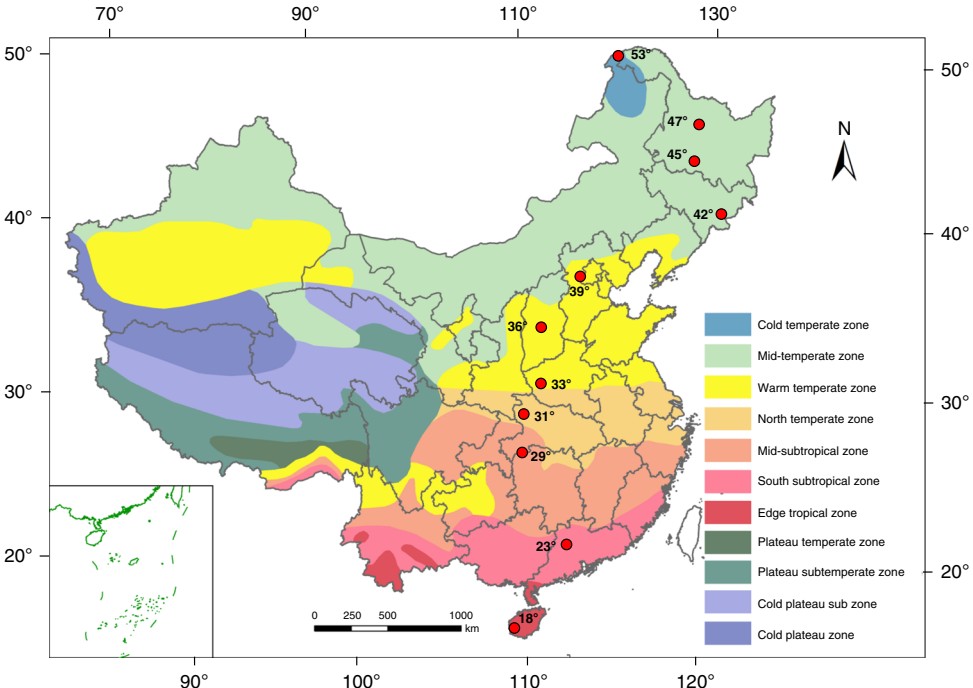

**Fig. 1 The distribution of 11 sampling regions along the latitudinal gradient transect.** Red circles indicate the sampling regions. In each region, 60 plots (each of 20 × 20 m) were established. A detail spatial distribution map of the sampling plots is shown in Supplementary Fig. 1. The source data are provided as a Source Data file.

gradient associated with many major forest types[24,25]. This large environmental gradient represents an ideal model to explore large-scale geographical patterns of β diversity. Also, a similar spatial arrangement of sampling plots among 11 regions is designed, allowing us to quantitively ascertain the mechanisms underlying the β diversity pattern associated with variation in regional species pools and local community assembly processes. Our study focuses on investigating β diversity patterns along the latitudinal gradient and how regional species pools and local community assembly processes mediate these patterns (as shown in Fig. 2) by addressing the following questions:

1. Is there a latitudinal gradient in β diversity of soil bacteria in eastern China? Earlier studies indicate that the turnover of bacterial communities (β diversity) is highly correlated with the variation in soil variables[13,23]. Considering a large variability and complexity of soil factors along the latitudinal gradient, we hypothesize that the observed β diversity pattern will not exhibit a latitudinal gradient.

2. What mechanisms underlie variation in β diversity along the latitudinal gradient? If variation in β diversity among different regions along the latitudinal gradient is caused simply by variation in the regional species pool, regardless of differences in local environmental constraints, we expect variation in β diversity to correlate with the sizes of species pools (γ diversity). Alternatively, local community assembly processes can significantly influence variation in β diversity. In this situation, the regional species pool itself is not a key driver of variation in β diversity, but variation in β diversity is due to differences in local community assembly processes.

3. If local community assembly processes explain differences in β diversity, how do local community assembly processes influence β diversity along the latitudinal gradient? Environmental selections or dispersal processes may cause variation in soil bacterial community composition[12,13]. Thus, we expect that heterogeneous selection or dispersal limitation will result in high β diversity and that homogeneous selection or homogenizing dispersal may lead to low β diversity.

We compare the effect of assembly mechanisms on β diversity pattern of soil bacteria along a latitudinal gradient with highly divergent species pools. Our results show that β diversity does not co-vary with γ diversity, and local community assembly mechanisms contribute to variation in β diversity patterns. Variation in β diversity along the latitudinal gradient is related to environmental heterogeneity, and variation in environmental conditions accounts for the variation in β diversity through mediating the strength of heterogeneous selection.

## Results

**Diversity patterns of bacterial community**. Relationships between soil bacterial (α, β, and γ) diversity and latitude all tended to be curvilinear, but greatly different (Fig. 3 and Supplementary Figs. 2 and 3). The distribution of α diversity along the latitudinal gradient was unimodal, with the highest diversity observed at 36–45° N (Fig. 3a, Supplementary Table 1). By contrast, low levels of β diversity were observed at 36–45° N and 23° N (Fig. 3b and Supplementary Table 1) and high levels of β diversity were found at 18°, 29–33°, 47°, and 53° N. The lowest γ diversity was observed at 23° N, followed by at 18° (tropical region) and at 53° N (cold temperate region), respectively, and the highest occurred at 31° N (Fig. 3c).

**The relation between β diversity and γ diversity**. To account for the effects of variation in regional species pool on β diversity, we

explored the expected algebraic relation between β diversity and γ diversity with a lognormal species abundance distribution in the absence of any process other than random sampling. We found that the expected β diversity increased with increasing γ diversity and that this expected relation between β diversity and γ diversity held regardless of variation in the number of individuals (Fig. 4a). However, for the soil bacterial communities across the latitudinal gradient, the correlation between observed β diversity and γ diversity was not consistent with the pattern expected (Fig. 4b and Supplementary Fig. 4), and no significant correlation was found between observed β diversity and γ diversity ($P = 0.25$, Fig. 4b and Supplementary Fig. 4).

**Community assembly processes**. Our results showed that the overall variation pattern of local community assembly (β deviation) along the latitudinal gradient was similar after controlling for variation either in γ diversity or in regional species pool (Fig. 5). After controlling for variation in regional species pool, all the mean β deviations were positive in 11 regions (significantly greater than zero, Supplementary Table 2). However, the mean β deviations at 18°, 29–33°, 47° and 53° N were significantly higher than at 23° and 36–45° N (Fig. 5d), indicating that the strength of local community assembly (i.e., environmental selection or dispersal limitation) significantly varied along the latitudinal gradient. In addition, variation in β deviations was positively correlated with observed β diversities along the latitudinal gradient after accounting for variation either in γ diversity or in regional species pool (Supplementary Fig. 5).

**Environmental and spatial factors affect community assembly**. To disentangle the influence of environmental selection and dispersal limitation on community assembly processes, we examined the extent to which β deviations varied across environmental and spatial gradients along the latitudinal gradient. In total, environmental, spatial variables and their joint effects accounted for 39.1–61.9% of the β deviations in the regions at 18°, 29–33°, 47°, and 53° N, but only explained 22.3–30.8% of the β deviations at 23° N and 36–45° N (Fig. 6). Across all these regions, environmental variables explained a larger fraction of the β deviations than spatial variables (Fig. 6). Among these environmental variables, soil organic carbon and nitrogen best explained the β deviations (Supplementary Table 6). Environmental variables explained a larger fraction of the β deviations at 18°, 29–33°, 47°, and 53° N than at 36–45° N (Fig. 6), and accordingly, the differences in environmental conditions were greater at the former regions than in the latter regions (Fig. 7a and Supplementary Table 7). Further, variation in β diversity along the latitudinal gradient was related with environmental heterogeneity (Fig. 7 and Supplementary Fig. 6).

## Discussion

There is increasing evidence that microorganisms and macro-organisms exhibit different biogeographic patterns along latitudinal gradients[2,6,26], but how soil bacterial β diversity varies with the latitudes, as well as its underlying mechanisms are unknown. Although variation in β diversity is often hypothesized to reflect the relative importance of different community assembly mechanisms[11,13], species pool itself may also drive variation in β diversity at the latitudinal scale[10,11,16]. Our study is to explore β diversity patterns along a latitudinal gradient transect and further disentangle the effect of local community assembly on β diversity from variation in regional species pools.

In our study, distinct distribution patterns of soil bacteria were observed with α, γ, and β diversity. Bacterial α diversity (local species richness) and γ diversity (regional species richness)

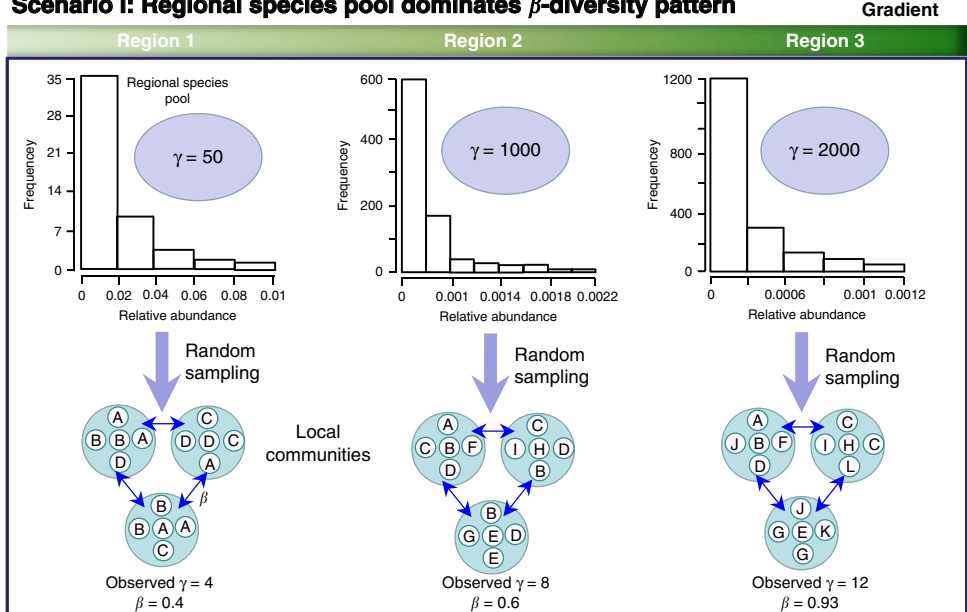

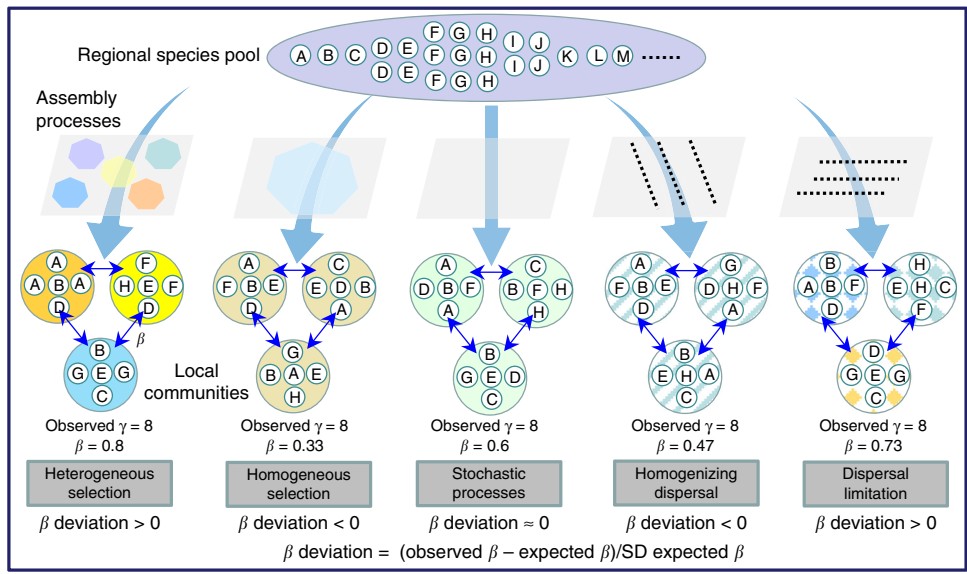

**Fig. 2 Hypotheses tested to illuminate the underlying drivers of bacterial β diversity patterns.** Scenario I: If regional species pools dominate β diversity pattern, then β diversity will vary depending on γ diversity and the distribution of species abundance. For example, with the increasing γ diversity along the latitudinal gradient, β diversity increases accordingly. In this study, γ diversity refers to regional species richness (a size of species pool), while regional species pool reflects both regional species richness and species abundance distribution. Scenario II: If local community assembly processes dominate β diversity pattern, the variation in β diversity mainly depends on local community assembly processes. For instance, β diversity is different due to different community assembly processes even when the observed γ diversity is consistent. β deviation is calculated as the difference between the observed and mean expected β diversity, divided by the standard deviation (SD) of expected values. Throughout the figure, letters represent hypothetical species. Large ovals represent regional species pools and small circles with letters represent local communities that comprise a subset of species from the regional species pool. The arrows going from the species pool to the local communities in scenario I reflect random sampling. The arrows passing through each respective condition in scenario II represent heterogeneous selection, homogeneous selection, stochastic processes, homogenizing dispersal and dispersal limitation, respectively. Here, stochastic processes refer to the ecological processes that give rise to β diversity patterns that are indistinguishable from random chance alone, such as ecological drift[35]. The shaded areas containing several hexagons represent environmental conditions, where hexagons with different colors represent environmental heterogeneity; a hexagon of a single color represents homogeneity. Dash lines perpendicular to the direction of arrows in the shaded area represent the dispersal limitation; dash lines parallel to the direction of arrows represent homogenizing dispersal.

exhibited a unimodal distribution (Fig. 3a, c and Supplementary Fig. 3a, c), which aligns with the global general pattern of soil bacteria[3], where bacterial species richness peaks at mid-latitudes and declines towards the poles and the equator. As expected, β diversity of soil bacteria did not exhibit a latitudinal gradient (Fig. 3b and Supplementary Fig. 2b). The reasons for the lack of an apparent latitudinal gradient in β diversity are manifold, among which is that diversity and abundance of bacterial communities are constrained primarily by edaphic variables[2,7,13]. Our results also showed that variation in β diversity along the

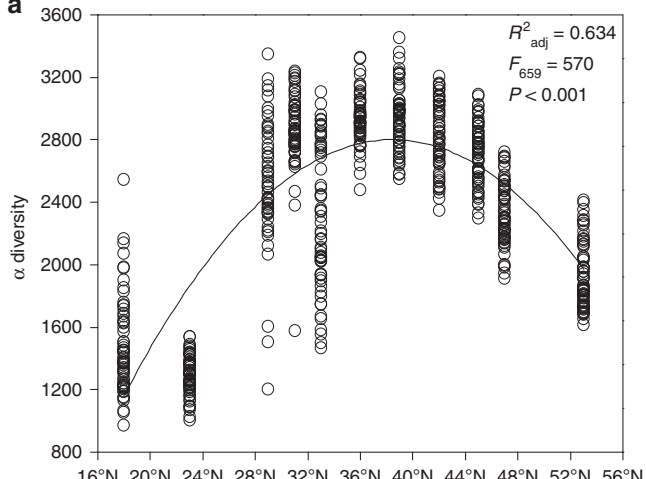

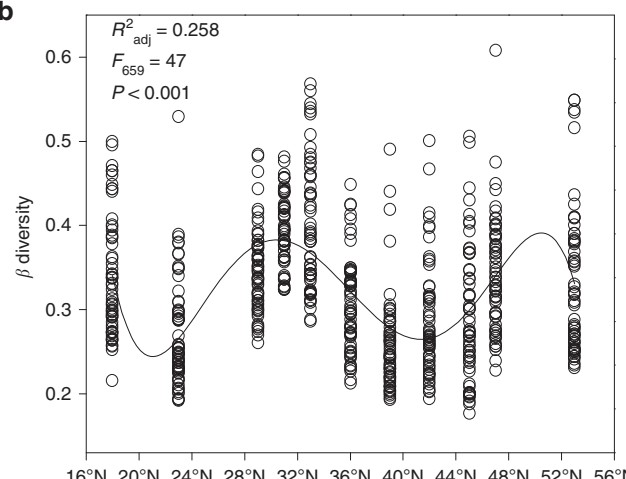

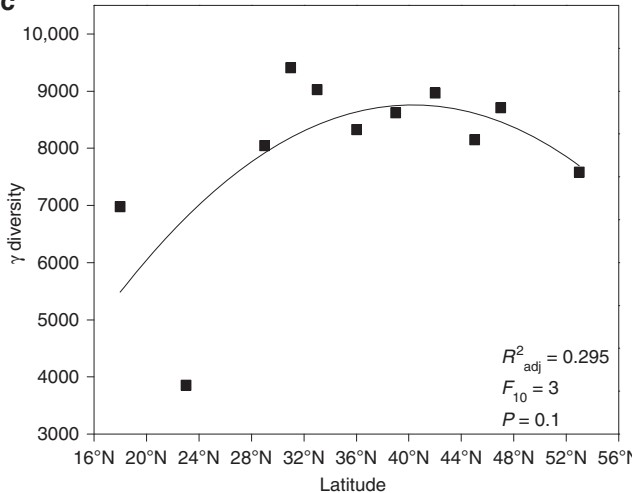

**Fig. 3 Soil bacterial diversity patterns along the latitudinal gradient. a** α diversity. **b** β diversity. **c** γ diversity. The relations between (α, β, and γ) diversity and latitude were evaluated using linear and polynomial regressions and the best polynomial fit was determined on the basis of Akaike information criterion (AIC) and the corrected Akaike information criterion AIC (AICc). The P-values by two-sided F-test are indicated in **a–c** (n = 60 independent samples). The source data are provided as a Source data file.

latitudinal gradient was significantly correlated with the heterogeneity of soil parameters (Fig. 7b), which is consistent with other studies showing that the turnover of soil bacterial composition is driven by changes in soil conditions[17,23]. These findings indicate that latitude, as a complex gradient along which many environmental variables are changing, is not sufficient for inferring large-scale controls on geographical patterns of bacterial diversity.

To identify the effects of regional species pool on β diversity, we first compared the expected relationship between β diversity and γ diversity in the absence of any process other than random sampling with the actual observed relationship in soil bacterial data using the method proposed by Kraft et al.[10]. The expected relation that β diversity increased with increasing γ diversity was not found in the observed β diversity measurements and no significant correlation existed between observed β diversity and γ diversity (Fig. 4 and Supplementary Fig. 4), suggesting that species pool richness did not dominate the variation in β diversity pattern. After correcting for differences in regional species pool, the differences in β diversity still existed and local community assembly processes appeared to explain the variation in β diversity along the latitudinal gradient (Fig. 5). Specifically, strong heterogeneous selection or dispersal limitation (as indicated by high β deviations at 18°, 29–33°, 47°, and 53° N, Fig. 5d) could increase β diversity by filtering species from the species pool in different ways, whereas weak heterogeneous selection or dispersal limitation (as indicated by low β deviations at the 23° and 36~45° N, Fig. 5d) may lead to relatively low β diversity. These results support the observed horseshoe effect[27] on the PCoA of bacterial community composition (Supplementary Fig. 7) and the results of Bahram (2018)[3], suggesting that soil bacterial communities at low and high latitudes are subjected to stronger environmental filtering and include a relatively greater proportion of edaphic-niche specialists, relative to bacterial communities at middle latitudes. Therefore, our study provides direct evidences that local community assembly mechanisms, not the regional species pool, shape soil bacterial β diversity pattern along the latitudinal gradient in eastern China. It is noted that β diversity is influenced by local environmental factors while could be likely to affect regional species pool as well. Further study is needed to better address the effects of β diversity on regional species pool.

Biogeographical variation in β diversity across environmental gradients may be strongly influenced by the degree of environmental heterogeneity[17,18]. In our study, the variation in β diversity was influenced by the strength of selection processes resulting from environmental heterogeneity (Figs. 5–7). The large variation in environmental conditions increased β diversity through enhancing heterogeneous selection at 18°, 29–33°, 47° and 53° N (Figs. 5d and 7). By contrast, the minor variation in environmental conditions at 36–45° N reduced the strength of heterogeneous selection, resulting in low β diversity (Figs. 5d and 7). This is consistent with recent studies that suggest environmental heterogeneity causes differences in community composition and β diversity in soil bacterial communities[17,23]. Further, our analyses suggest that the strength of selection is likely related to spatial heterogeneity in soil organic carbon and nitrogen (Supplementary Table 6), which is supported by several studies addressing important effects of organic carbon and nitrogen on soil bacterial communities[7,28]. For example, high heterogeneity in soil nitrogen can drive dissimilarity of bacteria communities by deliberately specific selection for copiotrophic or oligotrophic bacteria, which subsequently promote high soil bacterial β diversity[29,30]. However, the case of the 23° N region in this study may be an exception that relatively weak heterogeneous selection dominates bacterial community assembly with large variation in soil nitrogen (Figs. 5d and 7 and Supplementary Table 7). One possible explanation is that variation in soil nutrients does not

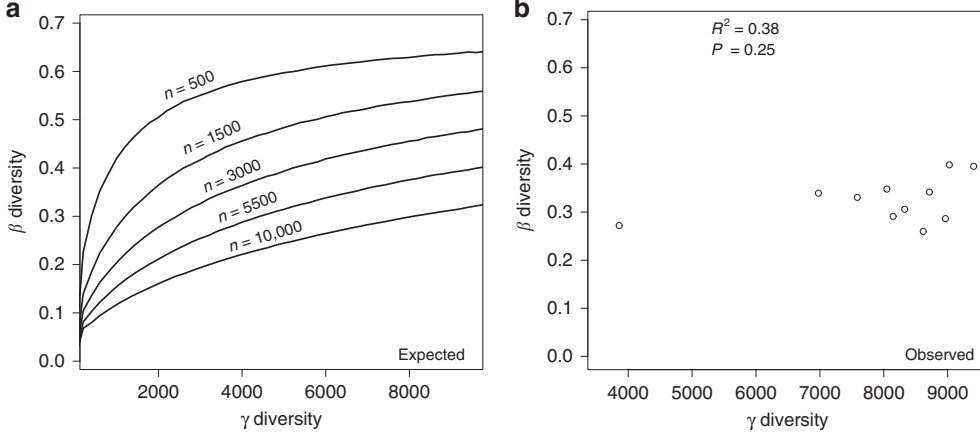

**Fig. 4 The relation between β and γ diversity. a** The expected algebraic relation between β and γ diversity produced by a simple sampling simulation, using a lognormal species abundance distribution. Curves represent β diversity values and 'n' represents the number of individuals in each plot. See Supplementary Fig. 4 for simulations showing similar relations for samples generated from uniform abundance distributions. **b** The observed relation between β and γ diversity in empirical data from soil bacterial communities along the latitudinal gradient. The *P*-values by two-sided spearman correlation are indicated in Fig. 4b. The source data are provided as a Source data file.

dominate community assembly processes under the extreme soil pH conditions[18,31]. The low soil pH observed in at 23° N region (DHS) could narrow the niche of soil bacterial communities, and consequently leads to low β diversity by selection of relatively similar species (Fig. 3b and Supplementary Table 7). Besides heterogeneous selection, high β diversity could also result from dispersal limitation. Therefore, it is necessary to evaluate the strength of each processes' specific effects on community assembly processes. Our results suggest that dispersal limitation has an important role in influencing β deviation, as explained by space effects. Furthermore, the explained variation was larger at 18°, 29–33°, 47° and 53° N than those at 36–45°N (Fig. 6). We attribute this response to the greater environmental heterogeneity in the former regions, which could decrease the establishment success of dispersed species and then reduce the similarity of bacterial community composition, leading to higher β diversity[21]. It is worth noting that these spatial effects could be the result of other underlying mechanisms that occur in parallel with dispersal limitation[8,32]. For example, unobserved spatially autocorrelated abiotic or biotic factors along the latitudinal gradient in this study could contribute to the differences in bacterial community composition[33]. In addition, historical processes, such as priority effects, where the first to arrive is able to exclude later colonists, could also further strengthen the effects of distance effects[34]. Further, uncaptured within-plot variability could increase the uncertainty in estimating the relationships between bacteria and the spatial or environmental distance[8]. Unexplained variation in β deviations reflects a combined effect of variation in species pools and unmeasured environmental and biological variables[35–37]. Especially, we found that a large portion (38.1–77.7%) of the variation in β deviation was unexplained by environmental and spatial factors. This is consistent with previous studies[13,38] in which 55.2–73% of the variation in soil microbial community composition and diversity was unexplained. Interestingly, we also found that the unexplained portion was greater in regions with relatively low environmental heterogeneity (36–45° N) than in regions with high environmental heterogeneity (18°, 29–33°, and 47° and 53° N). This may be due to unmeasured environmental variables and ecological processes, such as species competition for the same or similar resources, may contribute to more important part in low heterogeneous environment and β diversity[1,39,40]. In addition, given that our study is a snap-shot study, it is also possible that our findings may vary with time and season[41].

Collectively, these results emphasize the need for studies on multiple processes in theoretical ecological models.

In conclusion, local community assembly mechanisms, rather than regional species pool, dominate soil bacterial β diversity pattern along the latitudinal gradient transect. The variation in the strength of heterogeneous selection further improve understanding of how local community assembly processes influence β diversity pattern along the latitudinal gradient. Here, variation in environmental conditions leads to the variation in β diversity through mediating the strength of heterogeneous selection. Our study provides direct evidence illustrating the importance of local community assembly processes in shaping geographical patterns of soil bacterial β diversity along the latitudinal gradient transect.

## Methods

**Study sites and experimental design**. This study was conducted in natural forests along the latitudinal gradient transect, which spans 3,700 km with a latitudinal range from 53°47′ N to 18°69′ N and a longitudinal range from 108°87′ E to 128°91′ E (Fig. 1). The mean annual temperature (MAT) of the transect ranges from −4.4 to 21.9 °C. The mean annual precipitation (MAP) is from 482 to 2049 mm. Based on regional climates and geographic distribution, these sites were categorized into the seven climatic zones[42].

To elucidate the mechanisms underlying β diversity pattern, 11 regions were selected from north to south along the latitudinal gradient transect (Fig. 1 and Supplementary Fig. 1). In each region, 60 plots (each of 20 × 20 m²) were established randomly in a typical forest without anthropogenic or natural disturbance. In all the regions, the plots span similar spatial distances. The minimum spatial distance between the adjacent plots was 100 m and the maximum spatial distance between the plots in each region ranged between 8 and 11 km. The sampling plots in each region covered an area ranging from 22 to 30 km². The spatial arrangement of sampling plots was also similar across the 11 regions, thereby allowing us to compare the potential influence of environmental and spatial processes on β diversity.

Sampling occurred during June and July in 2015, starting from the south to the north along the transect. Soil samples were collected from all 660 plots using a uniform sampling protocol. Each sample was a composite of six individual soil cores (2.5 cm diameter × 10 cm depth) randomly collected from the horizon within each plot. Before sampling in each plot, the soil corer was cleaned and disinfected to prevent sample contamination. Each soil sample was transferred to a disposable sterile bag and all the samples were taken to the laboratory immediately in coolers. Discrete plant residues were manually removed from samples, and the soils were sieved through 2-mm plastic mesh and thoroughly homogenized. Each sample was divided into two parts: one was stored at 4 °C for soil property measurements and the other at –40 °C for DNA extraction.

**Environmental factors analysis**. Soil pH was measured in a soil-water slurry (1:5, w/v) using a pH meter. Total organic carbon (TOC) was quantified with the dichromate digestion method[43]. Total nitrogen (TN) was determined by combustion (CN-2000)[44]. Available nitrogen (AN) was measured by alkaline

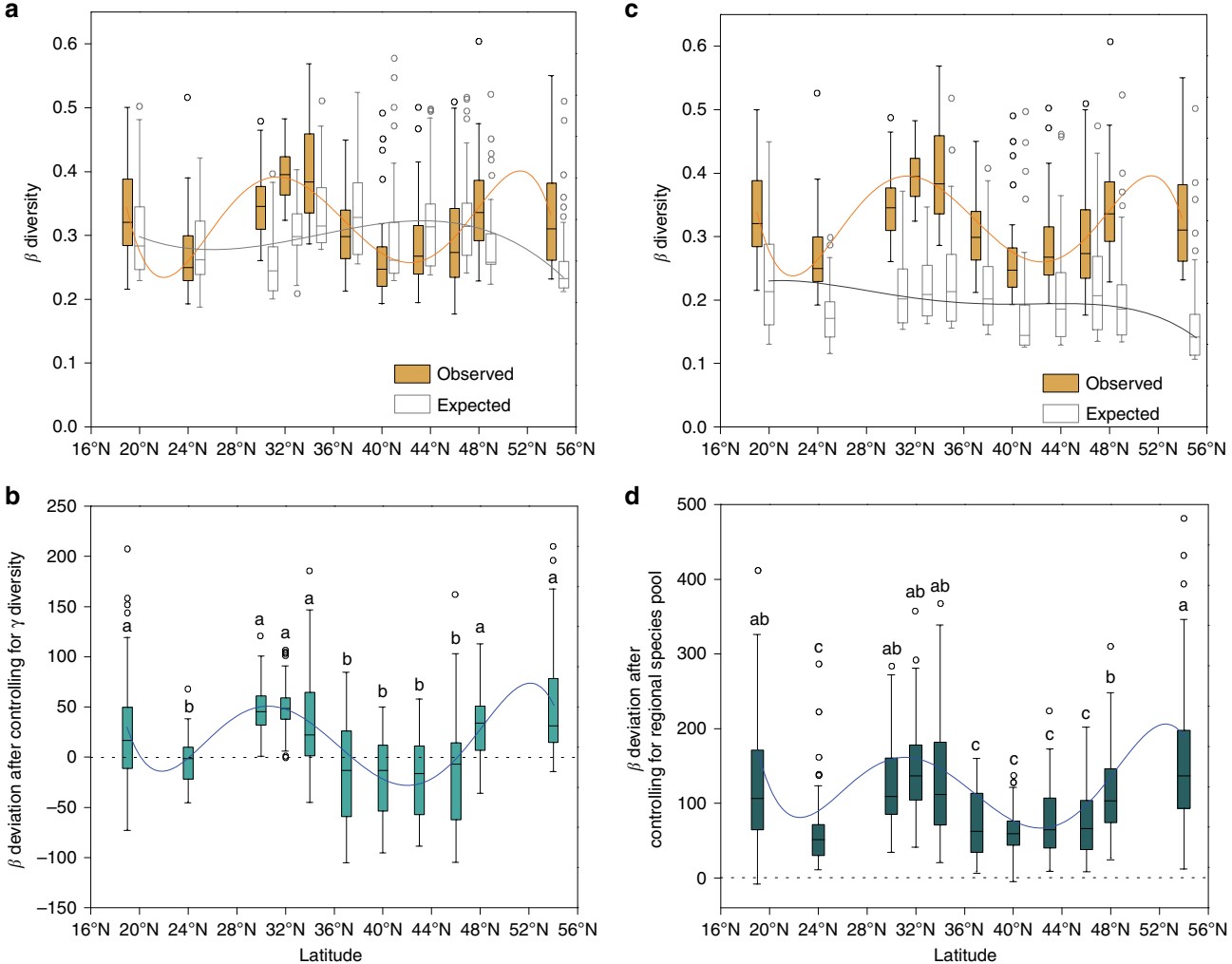

**Fig. 5 Patterns in β diversities and β deviations of soil bacteria along latitude. a** The observed (orange) and expected (gray) β diversities from the null model that controls for γ diversity; the observed β diversities were statistically different from expected β diversity for 76.7–98.3% samples across different regions (Supplementary Table 3). **b** Mean β deviations after controlling for γ diversity significantly varied along latitude (Supplementary Table 4). **c** The observed (orange) and expected (gray) β diversities from the null model that controls for regional species pool; the observed β diversities were statistically different from expected β diversity for 98.3–100% samples (Supplementary Table 5). **d** β deviations after controlling for regional species pool. The relations between β diversity (or β deviations) and latitude were evaluated using linear and polynomial regressions and the best polynomial fit was determined on the basis of Akaike information criterion (AIC). Letters indicate different regions that differ significantly at $P < 0.05$ in (**b**, **d**) (one-way ANOVA followed by multiple comparisons using Tukey HSD test, $n = 60$ independent samples). No adjustments were made for multiple comparisons. The bottom and top of each box represents the first and third quartiles, and the line inside is the median. The whiskers correspond to 1.5 times the interquartile range, while data beyond the whiskers are outlying points that are plotted individually. The source data are provided as a Source data file.

permanganate method[45]. Soil moisture (SM) was determined by the gravimetric method after the soil samples were oven-dried at 60 °C for 48 h. Climate attributes, including the mean annual temperature, mean annual precipitation and mean temperature of warmest quarter and coldest quarter of each region, were obtained from the WorldClim database (www.worldclim.org).

**Ribosomal RNA gene amplicon sequencing and processing**. Total microbial genomic DNA was extracted from 0.3 g of well homogenized soil for each sample using the MoBio PowerSoil DNA isolation kit (MoBio Laboratories, Carlsbad, CA, USA) according to the manufacturer's protocol. The quality and concentrations of the extracted DNA were assessed based on 260/280 nm and 260/230 nm absorbance ratios obtained using a NanoDrop Spectrophotometer (NanoDrop Technologies Inc., Wilmington, DE, USA). During the process of DNA extraction, PCR amplification and MiSeq sequencing, the strictest aseptic conditions were ensured to prevent samples contamination. To determine soil bacterial community composition and diversity, the V4 − V5 hypervariable regions of bacterial 16S rRNA genes were amplified by universal primer pairs, 515F (5′-GTGY-CAGCMGCCGCGGTA-3′) and 909R (5′-CCCCGYCAATTCMTTTRAGT-3′)[46]. The 5′end of the 515F primer included unique barcodes, which were later used to separate sequences by each sample. Each 25 µl PCR reaction volume contained 1 µl of each primer (10 µM), 10 ng of template DNA and 0.5 units of Accuprime high-

fdelity Taq polymerase. Each sample was amplified in triplicate using the following conditions: 28 cycles of denaturation at 94 °C for 30 s, annealing at 55 °C for 45 s, and extension at 72 °C for 45 s, and with a final extension at 72 °C for 5 min. Triplicate PCR reactions were pooled, subjected to gel electrophoresis and purified using a Gel Extraction kit (Omega Bio-Tek). PCR product from each sample was pooled in equimolar concentrations for sequencing. The sequencing library was prepared using Truseq DNA PCR-Free Library Preparation Kits and sequenced at Illumina Miseq platform with 2 × 250 bp V2 Kits.

**Processing of sequencing data**. The DNA sequence data were processed using QIIME–Version 1.7.0 (http://qiime.org/)[47]. Sequences were assigned to each sample based on the unique barcodes and the following quality-control criteria were implemented: sequences of length <200 bp were removed; a minimum average quality score of 30 was allowed; a maximum number of primer mismatches allowed was 2; no errors in barcode were allowed; no ambiguous bases were allowed[48]. Chimeras and singletons were removed using the Uchime algorithm[49]. Sequences classified as "unassigned" and "archaea" were also removed. Prior to analysis, samples were rarefied to the same sequence depth (7000 bacterial sequences per sample) using the Perl script daisychopper.pl[50,51]. The resulting high quality sequences were clustered into operational taxonomic units (OTUs) at the 97% sequence similarity threshold using a closed-reference OTU picking protocol

in QIIME[52,53]. Finally, representative sequences from each OTU were selected for taxonomic assignment using QIIME's Ribosomal Database Project (RDP) 16S Classifier[54]. Although the rarefaction curves were not saturated for observed species, distinctions among different regions were observed (see Supplementary Fig. 8). In addition, β diversity can be sufficiently described and more importantly explained by common taxa than rare taxa[55]. Thus, we could compare the variation in β diversity among different regions based on this sequencing depth.

**Statistical analyses**. We calculated the variation in (α, β, and γ) diversity along the latitudinal gradient. Here, α diversity was measured as the OTUs richness of a single 20 × 20 plot, γ diversity as the total OTUs richness of the 60 plots in each sampling region, and β diversity as the dissimilarity[56] in species composition among the 60 plots in each region. The relations between (α, β, and γ) diversity and latitude were evaluated using linear and polynomial regressions. Akaike information criterion (AIC) was used to select and compare the goodness of fit of the candidate linear and polynomial regression for the relationship between (α and β) diversity and latitude. Adjusted Akaike information criterion with correction for

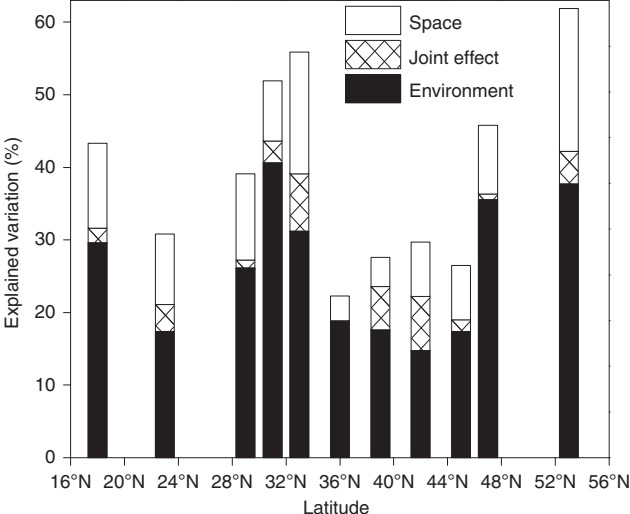

**Fig. 6 Percent of β deviations explained by environmental and spatial variables.** Environmental and spatial variables are listed in Supplementary Table 7. The source data are provided as a Source data file.

small sample size (AICc) was used to select the polynomial regression for γ diversity. Further, we examined the distribution of α and β diversity within each region using raw data and some potential outliers were found using the 1.5 interquartile range criteria (Supplementary Fig. 9). We also examined the distribution of γ diversity from 11 regions using raw data along the latitudinal gradient and a potential outlier was identified at 23° N region (Supplementary Fig. 10). In addition, the distribution of the mean α, β diversity and soil properties (pH, soil organic C, total N, and available N) from 11 regions were scrutinized, but no potential outlier was found (Supplementary Fig. 11). To assess the impact of the potential outlier, we conducted a sensitivity analysis comparing the model results with and without the potential outlier for all the relevant analyses including the regression analysis of (α, β, and γ) diversity (Fig. 3 and Supplementary Fig. 2 and 3), the correlation analysis between β diversity and γ diversity (Fig. 4 and Supplementary Fig. 4) and the correlation analysis between β diversity and environmental heterogeneity (Fig. 7 and Supplementary Fig. 6). Analysis results excluding the potential outlier were reported in the supplementary information and showed consistent conclusions in model estimations compared to the complete data analysis, although there were some differences in the curvilinear model of β diversity with (Fig. 3b) and without the outlier 23° region (Supplementary Fig. 3b). Throughout the manuscript, our analyses with the complete dataset are presented to ensure the high statistical power achieved by the robust experimental design.

To examine the effects of regional species pool on variation in β diversity, we first explored the intrinsic expected relationship between β diversity and γ diversity through a sampling simulation approach using a lognormal species abundance distribution and a uniform species abundance distribution (see R codes)[10]. In this way, the expected relation between β diversity and γ diversity in the absence of any community assembly process other than random sampling can be tested. Then, we examined the correlation between the observed β diversity and γ diversity along the latitudinal gradient using Spearman correlation. Here, if the observed relationship is consistent with the expected relationship, it suggests that variation in β diversity is primarily dependent on γ diversity. Conversely, an inconsistent relationship suggests γ diversity itself is not a dominant driver of variation in β diversity. In addition, null models were used to determine whether β deviations (that is, β diversity after controlling for variation in regional species pools) varied along the latitudinal gradient. If the magnitudes of β deviation varied systematically along the latitudinal gradient, it means that species pool does not dominate variation in β diversity.

To infer microbial community assembly processes, the method developed by Stegen et al.[12], the Raup-Crick metric[57,58] and β deviation (the null model method based on Bray-Cutis metric)[14,56,59] all were considered. Given that our study focused on comparing the relative importance of regional species pool and local assembly processes as primary drivers of β diversity, the β deviation method was chosen because it is commonly used to disentangle the effects of community assembly processes on β diversity from variation in species pool[56,59]. The effects of the regional species pool on β diversity consist of both γ diversity and regional species abundances distribution, and thus, in this study we examine how local community assembly processes influence β diversity using two different null

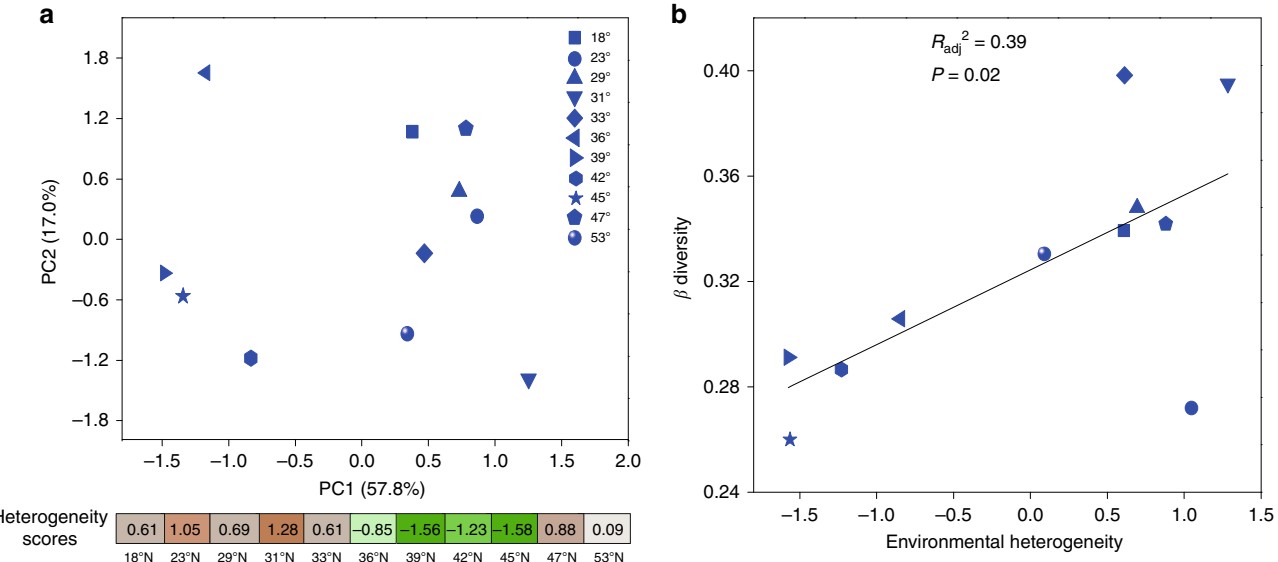

**Fig. 7 The relation between β diversity and overall environmental heterogeneity. a** Principal component analysis based on the coefficients of variation (CVs) of six environmental parameters (Supplementary Table 7). The heterogeneity score reflects the overall environmental variability in a region; a high score (orange color) represents a high heterogeneity; a low score (green color) represents a low heterogeneity. The heterogeneity score is a comprehensive and weighted number of the PC1 score, PC2 score and PC3 score. **b** The relation between β diversity and overall environmental heterogeneity was evaluated using linear regressions and the P-value by two-sided F-test in **b**. The source data are provided as a Source data file.

models that control for either one or both the two components of regional species pool[59]. In the first model, regional species pool is assumed as the observed number of species in a region. In this way, the regional species abundances are randomized by re-assigning individuals to each species in the region with equal probability. In the second null model, the species pool is defined as the observed number of species and abundances of species in a region. Here, the regional species abundance distribution is constrained to be the same in null and empirical datasets (see R code). Except for the difference in species pool, the calculation procedures of the two null models are the same. First, observed β diversity was calculated as the distance (or compositional dissimilarity) from an individual plot to the centroid of the group of all plots within a region (distance-to-centroid) using the multivariate method based on Bray-Curtis metric[56]. Then, random assemblages in each plot were generated from the species pool using the null model. Third, we calculated the β deviation as the magnitude that the observed β diversity deviates from the mean of expected β diversity (Fig. 2). We also statistically evaluated the difference between observed β diversity and the distribution of 999 expected β diversities for each sample (see R code, Source data, Supplementary Tables 3 and 5). In addition, one-sample t-test was used to compare mean β deviation against zero for 60 samples in each region and the P-values of the tests were presented (Supplementary Tables 2 and 4). A mean β deviation that is statistically indistinguishable from zero indicates the dominance of stochastic processes in shaping β diversity within a region; a mean β deviation significantly greater than zero indicates the dominance of heterogeneous selection or dispersal limitation in shaping β diversity; and a mean β deviation significantly less than zero indicates the dominance of homogeneous selection or homogenizing dispersal in shaping β diversity.

In this study, a range of environmental variables (Supplementary Table 8) from all the sampling regions were chosen, including: (1) five soil variables; (2) two topographic variables obtained from GIS or field measurements; and (3) four climatic variables. Spatial variables (Supplementary Table 9) included plot geographical coordinates (latitude and longitude) and spatial eigenfunctions were obtained from Principal Components of Neighbor Matrices (PCNM). To estimate the degree of spatial autocorrelation in a set of environmental variables, Moran's autocorrelation coefficient $I$[60] for each environmental factor was calculated. Across all regions, no spatial autocorrelation was observed for soil factors and slope, however, climatic variables and elevation showed significant spatial autocorrelation in most of the regions (Supplementary Table 8). The spatially autocorrelated variables were removed for the analyses on the effects of environmental factors on community assembly and β diversity. In the case of the collinearity among environmental variables, we removed variables that were highly correlated with other variables ($r \geq 0.80$; Supplementary Table 9). Then variables retained were used to partition the variation in the β deviations into individual fractions explained by environmental and spatial variables ("varpart" function in R vegan package). Forward-selection stepwise regression analysis ('step' function in the R stats package and Supplementary Data 1–4) was performed to examine the relationship between explanatory variables and β deviation[61] (Supplementary Table 6).

To further determine the relationship between environmental heterogeneity and β diversity, we first calculated the coefficients of variation (CVs) of six environmental parameters (excluding spatially autocorrelated variables) for each region (Supplementary Table 7). Then, principal component analysis (PCA) based on the CVs was used to reflect the variation in overall environmental heterogeneity across all the regions[41]. Finally, the relation between β diversity and environmental heterogeneity was statistically evaluated using linear regression.

**Reporting summary**. Further information on research design is available in the Nature Research Reporting Summary linked to this article.

## Data availability

The sequence data of bacterial community in this study are available in the NCBI Sequence Read Archive by accession no. PRJNA552986. Source data for Figs. 1, 3–7 and Supplementary Figs. 1–11 can be found in the source data file.

## Code availability

The R code supporting the findings presented here is available at https://github.com/YTHHN/R-code.

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

## Acknowledgements
We thank the following forest ecosystem research stations from CFERN and CERN, including Jianfengling, Dinghushan, Badagongshan, Shennongjia, Baotianman, Taiyueshan, Donglingshan, Changbaishan, Maoershan, Liangshui and Mohe, for their assistance in field sampling and data sharing. We also thank John Vance from West Virginia University and Alison Beamish from the University of British Columbia for their assistance with English language and grammatical editing of the manuscript, and a special appreciation goes to Prof. Dayong Zhang, from Beijing Normal University, Fangliang He, from University of Alberta, Liangdong Guo, from institute of Microbiolgy, Chinese Academy of Sciences, Osbert Sun, from Beijing Forestry University and Scott Chang, from University of Alberta, for their valuable advices on how to improve our manuscript. We gratefully acknowledge the joint supports of Natural Science Foundation of China (31930078, 31290223, CAFNSSFC201801 and 31700383) and Ministry of Science and Technology of China (2018YFC0507300), and China Biodiversity Observation Networks (Sino-BON).

## Author contributions
S.R.L. and X.Z. designed the study and conceived the paper. X.Z. sampled soil samples, Y.T.H. measured soil chemistry. X.Z., X.Z.L., M.J.Y., and X.J.L. were responsible of measures of bacterial community OTUs data and provided data on climates. K.L. was responsible for the options of data-processing and statistical methods. Z.F., S.J. were responsible for polishing the language of the paper. X.Z., S.R.L., J.X.W., Y.T.H., S.L.F., and H.W. wrote the first draft of the manuscript, and all authors substantially revised the paper.

## Competing interests
The authors declare no competing interests.
