## [Peer Review File · Nature Communications]

Reviewers' Comments:

Reviewer #1:

Remarks to the Author:

The MS by Zhang and co-authors considers the drivers of soil bacterial beta-diversity of 11 locations along a latitudinal transect. More precisely, they evaluate and compare the importance of the size of the local species pool (gamma-diversity) vs local assembly processes as primary drivers of beta-diversity. Despite the vast amount of microbial beta-diversity studies, the extent of the studied sites and the question makes the MS a valuable piece within the field.

However, the manuscript needs some improvement. First of all, I am missing references to some important works in the field. For example, the authors don't discuss or explain their choice of methods, and don't refer to existing frameworks. For example, I would be interested why community assembly wasn't analysed with the Stegen-method (Stegen et al., 2013)? Why did the authors choose the methods they used? Why didn't they use well-established metrics like the Raup-Crick metric to distinguish variable and homogeneous selection? These are well-established methods to assess microbial community assembly. It is okay not to use them but it would be preferable to explain why they don't use it.

Similarly, important microbial biogeography papers are missing, for example the work of Bahram et al (Nature, 2018) it's a piece that should be definitely discussed when evaluating the results of this study.

I am also missing the discussion of the potential limitations of this study. For example, the study is a snap-shot study and its findings may vary with time and season. Also, the potential of historical effects such as priority effects are not discussed.

Or the authors conclude that environmental heterogeneity influenced community assembly mechanisms but to what extent was the heterogeneity of the sites a result of the choice of the sampling? Were the plots chosen randomly? Was the environmental difference between the plots related to their distance (was there spatial autocorrelation? I don't think this was assessed)? I highly recommend to make available the exact location and map of the plots of each region in supplementary and analyze whether spatial autocorrelation was an issue for any of the environmental parameters.

Similarly, it has to be discussed that spatial effects in variance partitioning may emerge from unmeasured spatially autocorrelated parameters.

The language of the MS is quite good but there are some smaller grammatical mistakes. A few of them: l. 76-78, l. 109-110, Fig. figure legend.

Specific comments:

Figure 1: I think this figure is not really the best. Part a and b are okay but c and d are confusing. I also think it would be better to move this figure to the Methods part (l. 217-228).

l. 150: what do you mean by 'descending air temperature'?

l. 258: there is a very strong horseshoe effect on the PCoA. It could be discussed.

l. 346: I think your study only showed 'differences' and not 'changes' as it was a snap-shot study.

Figure 5: I had a hard time understanding these figures. They are missing a color legend, that has to be fixed. The figure is mentioned only once in l. 347, so I think it would be better to move it to supplementary. Instead a figure could be added about how overall environmental heterogeneity changed along the transect. To calculate environmental heterogeneity, see for example Langenheder et al, 2012.

l. 456-457: I don't agree that bacterial diversity changes with latitude would be 'the most striking

and frequently documented patterns in microbial ecology'. Please, tone down this claim.

Reviewer #2:

Remarks to the Author:

This study represents a unique dataset with a well-conceived design for testing hypotheses related to community assembly processes. The long latitudinal transect and the depth in replication at the regional scale is a great strength of this analysis and enables the researchers to control and account for many confounding factors. Furthermore, the major results of the analysis are important in supporting a fundamental tenet of community ecology that community assembly processes differ based on environmental heterogeneity. As such, latitude by itself is not sufficient for inferring large-scale controls on microbial diversity. These results warrant publication.

Numerous issues, however, remain to be resolved before this manuscript should be considered for publication. Here are my major and minor comments and suggestions.

- The manuscript is quite difficult to understand. While much of this may result from issues with translation as well as spelling and grammatical errors, there is also inconsistency in terminology. For example, the authors seem to refer to gamma-diversity interchangeably as 'species pools' or 'sizes of species pools' (Lines 100-103, 472), 'neutral sampling effects' (Lines 71-72), and 'sampling effects'. This ambiguity in language makes it difficult to understand and interpret the study. The authors should use consistent terminology throughout the manuscript.
- Given the confusing variety of jargon used in community ecology, it would be helpful to include an overview diagram of the different scales of diversity that also demonstrates how stochastic and deterministic processes coalesce to produce the expected beta-diversity outcomes. This could easily augment Figure 1.
- I would like the authors to consider what type of latitudinal gradient was expected? The manuscript is built around this key question however it is not adequately addressed. In addition, can the authors address whether the observed relationship between diversity and latitude was consistent with expectations or not?
- The authors did not systematically test the hypotheses and predictions outlined in Lines 100-118. Item #1, for example, lays out a clear prediction that 'If patterns of B-diversity are driven primarily by sampling effects, we expect changes of B-diversity to be coincided with the sizes of species pools (gamma diversity).' This analysis, however, does not appear in the manuscript. While other analyses are preformed, the appropriate analysis to test this hypothesis would be simply to determine the correlation between beta diversity to gamma diversity, as done in Kraft et al (ref 11). This would strengthen the argument on Lines 491-493.
- Lines 518-544: The authors must provide the raw data in order to evaluate the conclusions made in the text: what is a low pH? Also, if this site is an obvious outlier, including the site in the analysis is problematic. Finally, there is an apparent contradiction between Lines 534-536: "Our results differ by showing that soil pH has no effect on mediating the relative influences of stochastic and deterministic processes at local scales", and Lines 542-543: "we found a strong relationship between soil pH and local species pool (gamma diversity), indicating that soil pH plays an important role in influencing community assembly processes across latitudinal gradients...". The authors should correct this contradiction.
- Lines 491-493: I'm not convinced by the argument in this sentence that patterns in beta diversity differ between macro and micro-organisms for two reasons. First, there is only one study being refuted, even though the authors state in Line 491 stating that 'previous studies' demonstrated a different pattern in drivers of beta diversity in plants. Second, the referenced

study focused on woody plants. Given the large difference between the breadth of ecological niches occupied by woody plants compared to that of a complete soil microbial community, this may not be a fair comparison. If the authors can find additional support for this assertion it may be convincing. Otherwise, I recommend either removing or highlighting some of the major differences between this and the cited study.

- Lines 545-561: The argument for dispersal limitation is unconvincing. The spatial component of variance partitioning model can be used to infer dispersal limitation, however there is a very large amount of unexplained variation remaining after the variance partitioning analysis. Furthermore, the authors do not provide data on the spatial distribution of plots at each site, therefore the reader cannot determine how comparable each site is. This is, at best, a hypothesis to explain the spatial component, and the authors should be more cautious with their conclusion.

- The authors should consider reading and citing another relevant paper in the field: Martiny et al., Drivers of Bacterial B-Diversity Depend on Spatial Scale. PNAS, 2011.

Detailed comments:

- Please re-visit the english translation throughout the manuscript and the figure legends. There are numerous mis-spellings, odd word choices, and unclear sentences. For example, deviation is spelled 'devison' in the key to Figure 1 and as 'division' on line 345. There are many more examples. While each in itself would be relatively minor, they are frequent and create confusion when interpreting the manuscript.
- Lines 45-67: A simple overview diagram showing the interacting factors that contribute to B-diversity patterns would be helpful. The text by itself is difficult to read and understand.
- Figure 1: why is there a line connecting the symbols in the panels? It doesn't seem to have any meaning. I recommend removing.
- Throughout the manuscript, the authors switch between using singular and plural in the same sentence. Please be consistent.
- Methods: detail is generally lacking here. What is the study design? How far apart are the plots within a region? How large of an area do the plots in each region cover?
- Methods: the authors state that the study design accounts for spatial scale, however soil samples were pooled at the plot scale. How does this affect your results? Not being able to capture within-plot variability must increase the uncertainty in the relationships between microbes and the environment. This should be discussed as a potential study limitation
- Methods: Sampling and analysis details are also vague and make it difficult to interpret the validity of the methodology. Please also describe methods taken to keep samples free of contamination.
- Sequencing: Line 187, please cite the 909R primer; also, inconsistencies in sequencing method used: Line 189 indicates MiSeq, however Lines 194 and 196 indicate that pyrosequencing was used. Please rectify; Lines 198-203: Open or closed-reference OTU picking? More details on bioinformatics please.
- Results: Lines 253-259 and Figure S1, not sure why these are included? There is no analysis of the taxonomic data and it is not mentioned in the Discussion, therefore it seems superfluous.
- Figure 3: why is a polynomial being used to fit beta and gamma diversity? A unimodal relationship for alpha diversity is expected, however there's no clear justification for fitting the others using a polynomial function. Please support this decision.
- Lines 347, 349 and 350: Latitude values changed for 45 and 48 to 47 and 46, respectively. Please be consistent.
- Fig 5: Please add a color key, it's not clear how colors should be interpreted.
- Fig 6 legend: Bolivia and Missouri?!!
- Line 505: I believe Figure S2 is the incorrect figure to reference here.
- Please provide the environmental data in a separate table. It's impossible to interpret the author's statements about the homogenizing effect of pH on microbial communities at the 23N location (Lines 518-531) without seeing the pH data.
- Reference #46: Hubbell 2001: Please provide a chapter number.

Supplemental information:

- Table S1: The sample group information does not add significantly to the manuscript and I believe should be removed; why is there a footnote reading "Old growth with no sign of..." The footnote doesn't appear to be referenced in this Table. Please also include how far apart the plots are and how large the sites are.
- Table S2: First row has a 48 in it? Is this a typo? Please also add horizontal and vertical lines between rows and columns as they are difficult to read; Same comment applies to panel b); Please also explain why some check-marks appear in bold font?
- Fig S1: Fix typo.

Reviewer #1 (Remarks to the Author):

The MS by Zhang and co-authors considers the drivers of soil bacterial beta-diversity of 11 locations along a latitudinal transect. More precisely, they evaluate and compare the importance of the size of the local species pool (gamma-diversity) vs local assembly processes as primary drivers of beta-diversity. Despite the vast amount of microbial beta-diversity studies, the extent of the studied sites and the question makes the MS a valuable piece within the field.

However, the manuscript needs some improvement. First of all, I am missing references to some important works in the field. For example, the authors don't discuss or explain their choice of methods, and don't refer to existing frameworks. For example, I would be interested why community assembly wasn't analysed with the Stegen-method (Stegen et al., 2013)? Why did the authors choose the methods they used? Why didn't they use well-established metrics like the Raup-Crick metric to distinguish variable and homogeneous selection? These are well-established methods to assess microbial community assembly. It is okay not to use them but it would be preferable to explain why they don't use it.

Response: The reviewer raises an important point that we have inadequately and unclearly expressed in the original version. While, we noted and considered the Stegen-method (2013) and Raup-Crick metric, when choosing the methods for community assembly analysis in our study, as suggested, we provide a further explanation of method choice in the revised version, as described below. (Lines: 317-322)

To infer microbial community assembly processes, the methods developed by Stegen et al. (2013), Raup-Crick metric (Raup and Crick 1979, Chase et al. 2011) and β -deviation (the null model based on Bray–Curtis metric) (Ferrenberg et al. 2013, Vannette and Fukami 2017) all were considered. Given that our primary aims were to i) determine whether species pool or local community assembly mechanism dominates β -diversity patterns, and ii) to examine how community assembly processes influence patterns of β -diversity, the β -deviation was chosen for our study. The β -deviation is usually used to disentangle the effects of assembly processes on β -diversity from variation in species pool (Myers et al. 2015, Catano et al. 2017), and it is also a well-established method to assess microbial community assembly (Ferrenberg et al. 2013, Vannette and Fukami 2017).

Actually, β deviation in our study is similar to Raup-Crick metric. The Raup-Crick metric uses a null model to infer community assembly processes by comparing observed and expected β -diversity. Our method also uses the null model based on the Bray–Curtis metric to disentangle community assembly processes. The difference is that the Raup-Crick metric is usually used to disentangle variation in community dissimilarity from variation in α -diversity when communities are from the same species pool (Chase et al. 2011). In our study, β -deviation was used to disentangle community assembly from the effects of species pool, where species pools varied along the latitude.

Additionally, the Stegen method estimates microbial community assembly processes by quantifying both phylogenetic turnover and species turnover. It is worth noting that using phylogenetic turnover to infer ecological processes requires “phylogenetic signal” (that is, more closely related species have more similar ecological characteristics). However, we did not

find significant phylogenetic signal in all the locations in our study.

To better address the reviewer's questions and compare the differences in community assembly processes based on different methods, we also analyzed community assembly processes with the Stegen-method and found the main results were consistent with the findings based on the method we used. (Please see Figure 1 a and b).

Figure 1 (a) Patterns of β NTI across different locations along the latitude. Horizontal dashed lines indicate upper and lower significance thresholds at β NTI = +2 and -2, respectively. NTI > +2 indicates variable selection; NTI < -2 indicates homogeneous selection. (b) The percent of different community assembly processes (homogeneous, variable selection, dispersal limitation, drift and homogenizing dispersal) across different locations along the latitude.

Similarly, important microbial biogeography papers are missing, for example the work of Bahram et al (Nature, 2018). it's a piece that should be definitely discussed when evaluating the results of this study.

Response: We appreciate the reviewer's recommendation on the work of Bahram et al (Nature, 2018) to enrich the discussion of the results of this study.

We discussed that soil bacterial α diversity pattern with a unimodal distribution found in Bahram (2018) was consistent with the results of our study. (Lines 538-539)

Our study showed environmental heterogeneous and homogenous selection processes played an important role in mediating variations in β -diversity along the latitudinal gradient, which supported the results Bahram (2018) that soil bacterial communities at low and high latitudes were subjected to stronger environmental filtering and included a relatively greater proportion of edaphic-niche specialists, relative to at the middle latitudes. (Lines 558-561)

I am also missing the discussion of the potential limitations of this study. For example, the study is a snap-shot study and its findings may vary with time and season. Also, the potential of historical effects such as priority effects are not discussed.

Response: We agree with the reviewer's comments that our study is a snap-shot study and our findings may vary with time and season and we added this potential limitation in the Discussion (Lines 712-713). Also, space effects in distance-based redundancy analysis could be the influence

of other underlying mechanisms that can occur in parallel with dispersal limitation (Bell 2010, Martiny et al. 2011). For example, historical effects such as priority effects, where the first to arrive are able to exclude later colonists, could also further strengthen the effects of distance effects (Dexter et al. 2012). (Lines 678-688)

Or the authors conclude that environmental heterogeneity influenced community assembly mechanisms but to what extent was the heterogeneity of the sites a result of the choice of the sampling? Were the plots chosen randomly? Was the environmental difference between the plots related to their distance (was there spatial autocorrelation? I don't think this was assessed)? I highly recommend to make available the exact location and map of the plots of each region in supplementary and analyze whether spatial autocorrelation was an issue for any of the environmental parameters.

Response: We thank the reviewer's thoughtful comments and suggestions. In the revised manuscript, we added a detailed location description of sampling plots and provided their spatial distribution map in the supplementary (Supplementary Fig. 1) and checked spatial autocorrelation of the environmental parameters (Supplementary Table 1)

To elucidate the mechanisms underlying β -diversity pattern, 11 locations were selected from north to south along the latitudinal gradient in NSTEC (Fig. 2, Supplementary Fig. 1). In each location, 60 plots (each of $20 \times 20 \text{ m}^2$) were established randomly in a typical forest without anthropogenic or natural disturbance. In all the locations, the plots spanned similar spatial distances. The minimum spatial distance between the adjacent plots was 100 m and the maximum spatial distance between the plots in each location ranged between 8 and 11 km. In each location, the sampling plots cover an area ranging from 22 to 30 km^2 . The spatial arrangement of sampling plots was also similar among 11 locations, thereby allowing us to compare the potential influence of environmental and spatial processes on β -diversity. (Lines 187-198)

To estimate the degree of spatial autocorrelation in a set of environmental variables included in this study, Moran's autocorrelation coefficient I (Moran 1950) were calculated for each environmental factor. Across all the locations, no spatial autocorrelation was observed for soil factors and slope, while climatic variables and elevation showed significant spatial autocorrelation in most of the locations (Supplementary Table 1). Considering the autocorrelation between climate factors and spatial distance and collinearity among environmental variables, we removed variables that were highly correlated with other variables ($r \geq 0.80$; Supplementary Table 2a). Then, distance-based redundancy analysis (dbRDA) was used to disentangle their separate influences of environmental and spatial influences on community assembly. (Lines 341-369)

In addition, our results showed that soil factors accounted for variations in bacterial community assembly processes, whereas climate and topographical variables had little or no effect (Supplementary Table 2b). Therefore, spatial autocorrelation was not an issue for the environmental parameters influencing community assembly in our study.

Supplementary Figure 1 Geographical locations and spatial distribution of 20×20 m plots (red points) of 11 locations along the latitude in eastern China

Supplementary Table 1 Spatial autocorrelation for Mean annual Temperature (MAT), Mean annual precipitation (MAP), Mean Temperature of Warmest Quarter (MTWQ), Mean Temperature of coldest Quarter (MTCQ), Soil moisture, Soil pH, Soil organic carbon, Total nitrogen, Available nitrogen, Elevation and Slope.

location		MAT	MAP	MTWQ	MTCQ	SM	SOC	TN	AN	soil pH	Elevation	Slope
18°	Moran's I	0.76	0.69	0.75	0.77	-0.08	0.009	-0.09	0.1	-0.12	0.78	0.1
	P	< 0.001	< 0.001	< 0.001	< 0.001	0.81	0.36	0.88	0.09	0.9	< 0.001	0.08
23°	Moran's I	0.81	0.82	0.77	0.8	0.07	-0.01	-0.02	-0.01	0.86	0.48	0.09
	P	< 0.001	< 0.001	< 0.001	< 0.001	0.09	0.48	0.53	0.47	-0.09	< 0.001	0.12
29°	Moran's I	0.78	0.78	0.73	0.77	-0.17	0.1	-0.09	-0.04	-0.05	0.62	0.01
	P	< 0.001	< 0.001	< 0.001	< 0.001	0.98	0.89	0.85	0.63	0.71	< 0.001	0.31
31°	Moran's I	0.74	0.89	0.8	0.83	-0.08	0.06	-0.04	0.05	0.09	0.73	0.02
	P	< 0.001	< 0.001	< 0.001	< 0.001	0.81	0.13	0.65	0.19	0.1	< 0.001	0.32
33°	Moran's I	0.67	0.74	0.68	0.68	0.01	-0.16	-0.03	-0.09	-0.11	0.84	0.04
	P	< 0.001	< 0.001	< 0.001	< 0.001	0.32	0.98	0.6	0.85	0.9	< 0.001	0.2
36°	Moran's I	0.83	0.72	0.84	0.75	0.01	-0.03	-0.09	-0.02	-0.12	0.15	0.09
	P	< 0.001	< 0.001	< 0.001	< 0.001	0.33	0.59	0.84	0.53	0.92	0.002	0.56
39°	Moran's I	0.72	0.73	0.65	0.7	0.006	-0.1	0.04	-0.04	0.07	0.7	0.08
	P	< 0.001	< 0.001	< 0.001	< 0.001	0.37	0.89	0.21	0.66	0.1	< 0.001	0.09
42°	Moran's I	0.79	0.78	0.8	0.76	0.006	-0.06	-0.007	0.08	-0.1	0.83	0.07
	P	< 0.001	< 0.001	< 0.001	< 0.001	0.37	0.75	0.45	0.11	0.89	< 0.001	0.32
45°	Moran's I	0.68	0.76	0.7	0.66	-0.02	0.02	0.09	-0.12	-0.04	0.01	0.01
	P	< 0.001	< 0.001	< 0.001	< 0.001	0.51	0.32	0.06	0.09	0.67	0.33	0.06
47°	Moran's I	0.7	0.76	0.62	0.68	0.09	-0.05	-0.07	-0.01	-0.08	0.64	-0.1
	P	< 0.001	< 0.001	< 0.001	< 0.001	0.08	0.69	0.78	0.47	0.82	< 0.001	0.88
53°	Moran's I	0.14	0.09	0.13	0.12	-0.01	-0.14	-0.11	-0.06	-0.03	0.67	0.07
	P	0.003	0.03	0.003	0.006	0.47	0.96	0.89	0.72	0.6	< 0.001	0.35

Similarly, it has to be discussed that spatial effects in variance partitioning may emerge from unmeasured spatially autocorrelated parameters.

Response: We fully agree with the reviewer' comments. Spatial effects may emerge from unmeasured spatially autocorrelated parameters, and this was discussed in the revised manuscript, see Lines 678-682.

Lines 678-682: It is worth noting that these space effects could be the result of other underlying mechanisms that can occur in parallel with dispersal limitation (Bell 2010, Martiny et al. 2011). In our study, unobserved spatially autocorrelated abiotic or biotic factors along the latitudinal gradient could increase the differences in communities composition (Morlon et al. 2008).

The language of the MS is quite good but there are some smaller grammatical mistakes. A few of

them: l. 76-78, l. 109-110, Fig. figure legend.

Response: In the revised manuscript, grammatical mistakes were corrected. Also, we invited professional and native English speakers to further polish the writing language throughout the manuscript.

Specific comments:

Figure 1: I think this figure is not really the best. Part a and b are okay but c and d are confusing. I also think it would be better to move this figure to the Methods part (l. 217-228).

Response: We acknowledge the reviewer's helpful comment, and modified the Figure 1 (Page 6) in the revised manuscript that we hope the reviewer will find to be much improved.

Although we agree that original Figure 1 should be moved to Methods part, given that more information from INTRODUCTION is included in the updated Figure 1, we only modified the Figure 1.

l. 150: what do you mean by 'descending air temperature'?

Response: We would indicate that the North-south Transect of eastern China is characterized with a descending pattern of mean annual air temperature. Accordingly, the field sampling starting from the south to the north transect accompanied with a descending average annual air temperature. The statement was deleted in the revised manuscript for clarity.

l. 258: there is a very strong horseshoe effect on the PCoA. It could be discussed.

Response: The horseshoe-effect is suggestive of niche differentiation in microbial communities and indicates species turnover along environmental gradients (Morton, 2017). We discussed the very strong horseshoe effect on the PCoA, see Lines 558-564.

l. 346: I think your study only showed 'differences' and not 'changes' as it was a snap-shot study.

Response: We accepted and modified the expression accordingly as suggested, see Line 483.

Figure 5: I had a hard time understanding these figures. They are missing a color legend, that has to be fixed. The figure is mentioned only once in l. 347, so I think it would be better to move it to supplementary. Instead a figure could be added about how overall environmental heterogeneity changed along the transect. To calculate environmental heterogeneity, see for example Langenheder et al, 2012.

Response: We appreciate the reviewer's suggestion. We recalculated environmental heterogeneity according to the method in Langenheder et al (2012). (Page 19, the Figure 7)

As the updated Figure 7 clearly shows not only the variations in overall environmental heterogeneity, but also the coefficients of variation (CVs) of each environmental parameter (Supplementary Table 3), we delete the original Figure 5.

Our results indicate that higher total environmental heterogeneity occurred at 18°, 23°, 29-33°, 47°, and 53°N than that at 36~45°N (Fig. 7a), which is consistent with the original result. In addition, differences in β -diversity along the latitude were related with environmental heterogeneity (Fig. 7b).

Figure 7 (a) Principal component analysis based on the coefficients of variation (CVs) of 11 environmental parameters (Supplementary Table 3). The score reflects the overall environmental heterogeneity in each location, and a high score (red color) represents a high heterogeneity; a low score (green color) represents a low heterogeneity. The score of each location is a comprehensive and weighted number of the PC1 score, PC2 score and PC3 score. (b) The relation between β diversity and overall environmental heterogeneity.

I. 456-457: I don't agree that bacterial diversity changes with latitude would be 'the most striking and frequently documented patterns in microbial ecology'. Please, tone down this claim.

Response: We accepted the reviewer' comment and modified it as 'There are increasing evidences' instead of 'the most striking and frequently documented patterns in microbial ecology'. (Line 523)

Reviewer #2 (Remarks to the Author):

This study represents a unique dataset with a well-conceived design for testing hypotheses related to community assembly processes. The long latitudinal transect and the depth in replication at the regional scale is a great strength of this analysis and enables the researchers to control and account for many confounding factors. Furthermore, the major results of the analysis are important in supporting a fundamental tenet of community ecology that community assembly processes differ based on environmental heterogeneity. As such, latitude by itself is not sufficient for inferring large-scale controls on microbial diversity. These results warrant publication.

Numerous issues, however, remain to be resolved before this manuscript should be considered for publication. Here are my major and minor comments and suggestions.

The manuscript is quite difficult to understand. While much of this may result from issues with translation as well as spelling and grammatical errors, there is also inconsistency in terminology.

For example, the authors seem to refer to gamma-diversity interchangeably as ‘species pools’ or ‘sizes of species pools’ (Lines 100-103, 472), ‘neutral sampling effects’ (Lines 71-72), and ‘sampling effects’. This ambiguity in language makes it difficult to understand and interpret the study. The authors should use consistent terminology throughout the manuscript.

Response: We thank the reviewer’s positive comments on our manuscript. In the revised manuscript we were careful to use consistent terminology throughout. For example, we have referred to γ -diversity as ‘species pool’, rather than ‘sizes of species pools’ or ‘neutral sampling effects’. Also, we invited professional and native English speakers to check and polish the language, by correcting spelling and grammatical errors, and thus we believe, the revised version is greatly improved.

Given the confusing variety of jargon used in community ecology, it would be helpful to include an overview diagram of the different scales of diversity that also demonstrates how stochastic and deterministic processes coalesce to produce the expected beta-diversity outcomes. This could easily augment Figure 1.

Response: We appreciate the reviewer’s suggestion to augment Figure 1. As suggested, the revised figure 1 includes an overview diagram of the different scales of diversity that demonstrates how stochastic and deterministic processes coalesce to produce the expected β -diversity outcomes (Page 6, Figure 1).

Figure 1 Hypotheses tested in this study to illuminate the underlying drivers (i.e., species pool (γ -diversity) or local community assembly processes) of bacterial β -diversity patterns along a gradient. Scenario I: If species pool dominates β -diversity pattern, then β -diversity pattern will be dependent on variations in species pools. For example, with the increasing γ -diversity along the gradient, β -diversity increases accordingly. Scenario II: If local community assembly processes dominates β -diversity pattern, then the variations in β -diversity along a gradient will be distinct from the variation trend in species pool. For instance, as γ -diversity increases along the gradient,

β -diversity does not increase correspondingly. Throughout the figure, letters represent hypothetical species. Large ovals with letters represent species pools and small circles with letters represent local communities. Local communities comprise a subset of species from the species pool that have passed through the communities assembly processes (e.g. heterogeneous selection, homogeneous selection, dispersal limitation and stochastic process). Arrows represent the community assembly processes from species pools to local communities. The shaded areas containing several hexagons represent environmental conditions, where hexagons with different colors represent soil environmental heterogeneity; a single color represents soil homogeneity. Dash lines in the shaded area represent the dispersal limitation.

I would like the authors to consider what type of latitudinal gradient was expected? The manuscript is built around this key question however it is not adequately addressed. In addition, can the authors address whether the observed relationship between diversity and latitude was consistent with expectations or not?

Response: We appreciate the reviewer's thoughtful question and suggestion. In our study, we conducted a long latitudinal transect survey, which covers a great variety of climate types and a full spectrum of the environmental gradient along the North-South Transect of eastern China. This large gradient represents an ideal model to explore large-scale β -diversity pattern and test whether β -diversity exhibits the typical increasing diversity pattern from the boreal zone to the tropics, as is seen in many macroscopic organisms (Willig et al. 2003). Our study found that, variations in β diversity did not show a latitudinal gradient of increasing diversity from high latitude to low latitude. We considered expected latitudinal gradient (Lines 102-104) and addressed observed and expected diversity patterns in the discussion (Lines 540-542).

Even though few studies directly compare β -diversity latitudinal pattern with ours, some studies of beta diversity patterns across other environmental gradient support our findings. For example, studies both in the whole France (Ranjard et al. 2013) and the drylands of northern China (Wang et al. 2017) found that β -diversity varied with habitats due to difference of soil conditions. Considering there are different habitats for soil bacterial communities across the latitude (Bahram et al. 2018), together with variations in soil factors without latitudinal gradient (Post et al. 1985), it is quite probable that there is no significant influence of latitude on bacterial β -diversity which measures the dissimilarity of composition among different communities within a location.

More importantly, as the reviewer's comments, we conducted an unique design with 11 locations across a large spatial scale and with the similar spatial arrangement of sampling plots among different locations, allowing us to quantitatively ascertain the mechanisms underlying the β -diversity pattern associated with variations in species pools and local community assembly processes. Our study focused on distinguishing the relative importance of local community assembly mechanisms and species pool, in addition to β -diversity pattern. Thus, to be more focused on the core content, we did not discuss lots of detail about β -diversity pattern in the revised manuscript.

The authors did not systematically test the hypotheses and predictions outlined in Lines 100-118. Item #1, for example, lays out a clear prediction that 'If patterns of B-diversity are driven primarily by species pool, we expect changes of B-diversity to be coincided with the sizes of species pools (gamma diversity). This analysis, however, does not appear in the manuscript.

While other analyses are preformed, the appropriate analysis to test this hypothesis would be simply to determine the correlation between beta diversity to gamma diversity, as done in Kraft et al (ref 11). This would strengthen the argument on Lines 491-493.

Response: We thank the reviewer's comment and suggestions. We have addressed the hypotheses, 'If patterns of β -diversity are driven primarily by species pool, we expect changes of β -diversity to be coincided with the sizes of species pools (γ diversity)', as described in Kraft et al (2011).

The expected relation between β - and γ -diversity showed that β -diversity increased with species pool (γ diversity) (Fig. 4a). By contrast, this positive correlation was not found in observed β -diversity (Fig. 4b). There was no significant correlation between observed β -diversity and species pool ($R^2 = 0.32$, $P = 0.34$). These findings suggested that species pool did not dominate the variations in β -diversity along the latitude. We added these contents in the METHODS, RESULTS and DISCUSSION, respectively. (Lines 299-307, Lines 424-438 and Lines 545-551)

No significant relation between observed β -diversity and species pool was mutually confirmed by the result that differences in β -diversity still existed after correcting for the influences of species pools, which further strengthen our argument.

Figure 4 (a) The expected relation between β and γ diversity produced by a simple sampling simulation proposed in Kraft et al (2011). Curves represent β -diversity values, measured as the Bray-Curtis dissimilarity. ' n ' represents the number of individuals in each plot. (b) The relation between observed β - and γ -diversity in empirical data from soil bacterial communities along the latitudinal gradient. In this study, a potential outlier was identified when examining γ -diversity, which was outlined in the following response and Supplementary Fig. 2. We conducted a sensitivity analysis comparing the model results with and without the potential outlier for correlation analysis between β -diversity and species pool. Analysis results excluding this potential outlier were reported in the supplementary (Supplementary Fig. 5), and we found no difference in the model estimations and conclusions compared with the complete data analysis (for the complete data $R^2 = 0.32$, $P = 0.34$, for the subset data excluding this outlier $R^2 = 0.09$, $P = 0.8$).

Supplementary Figure 5 (a) The expected relation between β and γ diversity produced by a simple sampling simulation proposed in Kraft et al (2011). Curves represent β -diversity values, measured as the Bray-Curtis dissimilarity. ' n ' represents the number of individuals in each plot. (b) The relation between observed β - and γ -diversity in empirical data from soil bacterial communities (excluding the potential outlier at 23° N location) along the latitudinal gradient.

Lines 518-544: The authors must provide the raw data in order to evaluate the conclusions made in the text: what is a low pH? Also, if this site is an obvious outlier, including the site in the analysis is problematic. Finally, there is an apparent contradiction between Lines 534-536: “Our results differ by showing that soil pH has no effect on mediating the relative influences of stochastic and deterministic processes at local scales”, and Lines 542-543: “we found a strong relationship between soil pH and local species pool (gamma diversity), indicating that soil pH plays an important role in influencing community assembly processes across latitudinal gradients...”. The authors should correct this contradiction.

Response: In the modified manuscript, raw data are available and our contradictory arguments have been corrected.

First, the environmental raw data can be found in Supplementary Table 3 to support the conclusions drawn from this study.

Second, we examined whether the 23° N location is an obvious outlier or not by examining the distribution of bacterial (α -, β - and γ -) diversity and soil properties (pH, soil organic C, total N and available N). We found that the 23° location was not an outlier with soil pH, bacterial α and β diversity, but was identified to be a potential outlier with γ diversity using the 1.5 interquartile range criteria (Supplementary Fig. 2).

Third, To assess the impact of the potential outlier, we conducted a sensitivity analysis comparing the model results with and without the potential outlier for all the relevant analyses (the regression analysis of (α -, β - and γ -) diversity, the correlation analysis between β -diversity and γ -diversity and the correlation analysis between β -diversity and environmental heterogeneity). Analysis results excluding the potential outlier were reported in the supplementary data and showed no difference in model estimations and conclusions comparing to the complete data analysis. Therefore, we consider our analyses and conclusions to be robust and unbiased with regards to this potential outlier. Furthermore, we decided to maintain good statistical power and enough information by keeping all measured data. (Lines 288-298) (For this

question, we also asked a biostatistician from University of Toronto, for her professional advices on judgment of potential outlier.)

Finally, as for the “contradictory description”, we acknowledge and apologize that our original statement did not narrate our point precisely and clearly. In the revised manuscript, we modified the expression for clarity. Here, our results suggested that soil pH influences the assembly processes at the latitudinal scale, yet pH has little effect at a local scale (Supplementary Table 2).

Specifically, in our study, the smallest species pool was associated with the lowest pH at latitude 23° location. This is also in line with previous study that low bacterial phylotype diversity occurred with extreme acidic or alkaline pH, relative to neutral pH value (Tripathi et al. 2018). Moreover, we found a significant relationship between soil pH and species pool (γ diversity) (Supplementary Fig. 7), indicating that soil pH may play an important role in affecting community assembly processes at the latitudinal scale, leading to differences in species pool. However, our results also showed that soil pH had no effect on assembly processes at a local scale (that is, within each location). This may be due to soil pH being relatively consistent within a location, although there was a large variation range of soil pH (ranged from 3.98 to 6.71) across latitudinal gradients in the NSTEC.

However, given that our study focused on how species pool and local community assembly processes influence bacterial β pattern, we rephrased the discussion of soil pH results for a better logical structure. See Lines 620-626.

Lines 620-626: In our study, the low soil pH observed in at 23° N location narrows the niche of soil bacterial communities and consequently leads to small species pool. In fact, we also found that soil pH was related with species pool (Supplementary Fig. 7), which indicated pH influence latitudinal community assembly processes, leading to differences in species pool, although it did not seem to influence local community assembly processes (Supplementary Table 2b), especially heterogeneous selection .

Supplementary Table 3 Summary of environmental factors at different locations along the latitude. Mean values and coefficients of variations (CVs, %) calculated from 11 locations are shown for the following parameters: Mean annual Temperature (MAT), Mean annual precipitation (MAP), Mean Temperature of Warmest Quarter (MTWQ), Mean Temperature of coldest Quarter (MTCQ), Soil moisture (SM), Soil pH, Soil organic carbon (SOC), Total nitrogen (TN), Available nitrogen (AN), Elevation and Slope.

Locations		MAT (°C)	MAP (mm)	MTWQ (°C)	MTCQ (°C)	SM (%)	SOC (g/kg)	TN (g/kg)	AN (mg/kg)	soil pH	Elevation (m)	Slope (°)
18°	mean	21.5	1547.54	24.95	16.96	30.94	35.43	1.97	146.96	5.05	889.72	20.56
	CV	2.27	2.4	1.95	2.84	19.98	40.83	40.35	52.96	7.68	15.96	12.3
23°	mean	21.99	1594.11	28.19	14	34.1	41.8	1.85	104.79	3.98	270.43	20.8
	CV	1.82	1.42	1.52	2.44	19.07	39.97	56.36	50.25	8.08	16.71	15.44
29°	mean	12.61	1427.41	22.02	2.63	34.82	91.18	4.93	355.68	4.88	1394.42	26.01
	CV	7.43	1.27	4.61	10.37	17.52	48.74	47.17	43.04	11.82	13.15	12.79
31°	mean	9.98	1189.33	19.56	-0.21	35.05	74.39	4.66	573.71	6.12	1452.18	25.85
	CV	6.23	3.2	3.6	-7.68	16.62	38.14	49.33	61.61	9.33	11.34	19.76
33°	mean	11.02	819.36	22.14	-0.1	29.07	47.14	2.91	317.34	4.91	1076.18	27.93
	CV	6.14	2.13	3.41	-10.28	17.95	40.79	48.15	44.09	5.6	17.98	15.92
36°	mean	6.59	581.18	18.13	-6.64	23.73	41.42	2.71	261.85	6.28	1523.55	24.8
	CV	4.75	1.42	4.59	-3.75	21.33	26.48	22.36	21.65	3.6	15.89	12.06

39°	mean	6.27	495.93	20.06	-9.15	29.36	112.91	3.16	188.3	6.72	1319.08	25.1
	CV	2.85	0.52	0.93	-1.78	16.27	16.66	19.28	30.24	3.27	8.53	12.13
42°	mean	2.21	677.77	17.66	-15.23	36.06	95.55	7.01	502.35	5.29	814.62	7.08
	CV	13.86	1.15	1.89	-1.59	13.05	29.63	23.49	28.55	8.53	5.48	12.81
45°	mean	1.32	666.82	18.83	-18.78	41.39	67.18	5.84	439.09	5.69	458.23	17.48
	CV	10.12	0.31	0.54	-0.86	14.97	19.53	29.72	20.61	5.28	9.34	11.82
47°	mean	0.46	647.9	18.13	-20.79	43.7	120.59	6.82	412.1	5.81	378.08	14.85
	CV	6.64	0.37	2.48	-3.41	20.34	37.41	48.41	41.33	15.09	16.11	13.87
53°	mean	-4.23	423.57	16.67	-26.48	34.7	55.67	1.86	178.82	5.43	486.08	6.44
	CV	-3.5	0.66	2.04	-4.46	14.5	34.45	54.39	47.99	9.1	16.32	12.72

Supplementary Figure 2 Box-plot of bacterial (α -, β - and γ -) diversity and soil properties (pH, soil organic C, total N and available N)

Lines 491-493: I'm not convinced by the argument in this sentence that patterns in beta diversity differ between macro and micro-organisms for two reasons. First, there is only one study being refuted, even though the authors state in Line 491 stating that 'previous studies' demonstrated a different pattern in drivers of beta diversity in plants. Second, the referenced study focused on woody plants. Given the large difference between the breadth of ecological niches occupied by woody plants compared to that of a complete soil microbial community, this may not be a fair comparison. If the authors can find additional support for this assertion it may be convincing. Otherwise, I recommend either removing or highlighting some of the major differences between this and the cited study.

Response: We fully agree with the reviewer's comments and suggestions, and therefore, we deleted this assertion.

Lines 545-561: The argument for dispersal limitation is unconvincing. The spatial component of variance partitioning model can be used to infer dispersal limitation, however there is a very large amount of unexplained variation remaining after the variance partitioning analysis. Furthermore, the authors do not provide data on the spatial distribution of plots at each site, therefore the reader cannot determine how comparable each site is. This is, at best, a hypothesis to explain the spatial component, and the authors should be more cautious with their conclusion.

Response: We appreciate the reviewer's valuable comments that helped us to reframe the discussion and consider wider aspects of the spatial component in our variance partitioning model. In the revised discussion, we have added various possible explanations for dispersal

limitation by citing more relevant references (Lines 678-688). We provided the spatial distribution map of sampling plots for each location in the supplementary (Supplementary Fig. 1).

Lines 678-688: It is worth noting that these space effects could be the result of other underlying mechanisms that can occur in parallel with dispersal limitation (Bell 2010, Martiny et al. 2011). In our study, unobserved spatially autocorrelated abiotic or biotic factors along the latitudinal gradient could contribute to the differences in bacterial community composition (Morlon et al. 2008). Meanwhile, within-plot variability that our study did not capture could increase the uncertainty in estimating the relationships between bacteria and the spatial or environment distance (Martiny et al. 2011). Also, ecological drift, such as random fluctuations, stochastic reproduction and mortality, along with restricted movement of bacteria, could be involved to exacerbate dispersal limitation (Bahram et al. , Bell 2010). In addition, historical effects such as priority effects, where the first to arrive are able to exclude later colonists, could further strengthen the effects of distance effects(Dexter et al. 2012).

The authors should consider reading and citing another relevant paper in the field: Martiny et al., Drivers of Bacterial B-Diversity Depend on Spatial Scale. PNAS, 2011.

Response: We appreciate the suggestion, and have read and cited the paper (Martiny et al., 2011), see Line 42, Line 684.

Detailed comments:

- Please re-visit the english translation throughout the manuscript and the figure legends. There are numerous mis-spellings, odd word choices, and unclear sentences. For example, deviation is spelled 'devison' in the key to Figure 1 and as 'division' on line 345. There are many more examples. While each in itself would be relatively minor, they are frequent and create confusion when interpreting the manuscript.

Response: Again, we apologize for these spelling and grammatical errors. In this revised manuscript, we have carefully checked and corrected all typos and unclear sentences. We invited professional and native English speakers to check and polish the language a couple of times. Thus, we believe the revised version is much improved.

Lines 45-67: A simple overview diagram showing the interacting factors that contribute to B-diversity patterns would be helpful. The text by itself is difficult to read and understand.

Response: We thank the reviewer's suggestion, and a simple overview diagram was created to show the interacting factors that contribute to β -diversity patterns, please see Figure 1 on page 6.

Figure 1: why is there a line connecting the symbols in the panels? It doesn't seem to have any meaning. I recommend removing.

Response: As suggested, the line connecting the symbols in Fig 1 was removed.

Throughout the manuscript, the authors switch between using singular and plural in the same sentence. Please be consistent.

Response: We carefully checked and modified the manuscript, and kept singular or plural

consistent in the same sentence.

Methods: detail is generally lacking here. What is the study design? How far apart are the plots within a region? How large of an area do the plots in each region cover?

Response: We added more detailed information in the Methods as suggested. The “study design” was provided in the Methods, please see Lines 187-198. In addition, we provide spatial distribution maps of the plots at each location (Supplementary Fig. 1).

Lines 187-198: To elucidate the mechanisms underlying β -diversity pattern, 11 locations were selected from north to south along the latitudinal gradient in NSTEC (Fig. 2, Supplementary Fig. 1). In each location, 60 plots (each of $20 \times 20 \text{ m}^2$) were established randomly in a typical forest without anthropogenic or natural disturbance. In all the locations, the plots span similar spatial distances. The minimum spatial distance between the adjacent plots was 100 m and the maximum spatial distance between the plots in each location ranged between 8 and 11 km. In each location, the sampling plots cover an area ranging from 22 to 30 km^2 . The spatial arrangement of sampling plots was also similar among 11 locations, thereby allowing us to compare the potential influence of environmental and spatial processes on β -diversity.

Methods: the authors state that the study design accounts for spatial scale, however soil samples were pooled at the plot scale. How does this affect your results? Not being able to capture within-plot variability must increase the uncertainty in the relationships between microbes and the environment. This should be discussed as a potential study limitation

Response: We agree with the reviewer’s comments. In our study, pooled soil samples within a plot can increase the uncertainty in assessing the relationships between microbes and the environment because within-plot variability could not be fully captured. As suggested, its potential study limitation was discussed in our revised manuscript (see Lines 682-684).

Methods: Sampling and analysis details are also vague and make it difficult to interpret the validity of the methodology. Please also describe methods taken to keep samples free of contamination.

Response: We thank the reviewer’s comments and suggestions. We have added detail description of sampling and analysis, including the methods taken to keep samples free of contamination. See Lines 202-214, Line 236-255.

Lines 202-214: Sampling occurred during June and July in 2015, starting from the south to the north along the NSTEC. Soil samples were collected from 660 plots using a uniform sampling protocol. Each sample was a composite of six individual soil cores (2.5 cm diameter \times 10 cm depth) collected from the surface horizon within each plot. Soil collector was cleaned and disinfected before sampling in each plot. Each soil sample was transferred to a disposable sterile bag and all the samples were taken to the laboratory immediately in coolers. Discrete plant residues were manually removed from samples, and the soils were then sieved to pass through a 2-mm plastic mesh and homogenized thoroughly. Each sample was divided into two parts: one was stored at 4°C for soil property measurements and the other part at -40°C for DNA extraction.

L 236-255: Total microbial genomic DNA was extracted from 0.3 g of well homogenized soil for each sample using the MoBio PowerSoil DNA isolation kit (MoBio Laboratories, Carlsbad, CA, USA) according to the manufacturer's protocol. The quality and concentrations of the extracted DNA were assessed based on 260/280 nm and 260/230 nm absorbance ratios obtained using a NanoDrop Spectrophotometer (NanoDrop Technologies Inc., Wilmington, DE, USA). During the whole process of DNA extraction, the following PCR amplification and MiSeq sequencing, the strictest aseptic conditions were ensured to keep samples free of contamination. To determine soil bacterial community composition and diversity, the V4–V5 hypervariable regions of bacterial 16S rRNA genes was amplified by universal primer pairs, 515F (5' -GTGYCAGCMGCCGCGGTA-3) and 909R (5' -CCCGYCAATTCMTTTRAGT-3') (Tamaki et al. 2011). The 5' end of the 515F primer included unique barcodes, which were later used to split sequences by each sample. Each 25 μ l reaction volume contained 1 μ l of each primer (10 μ M), 10 ng of template DNA and 0.5 units of Accuprime high-fidelity Taq. Each sample was amplified in triplicate under the following conditions: 28 cycles of denaturation at 94° C for 30 s, annealing at 55° C for 45 s, and extension at 72° C for 45 s, and with a final extension at 72° C for 5 min. Triplicate PCR reactions were pooled and subjected to electrophoresis and purified using a Gel Extraction kit (Omega Bio-Tek). PCR product from each sample was pooled in equimolar concentrations for sequencing. The sequencing library was prepared using Truseq DNA PCR-Free Library Preparation Kits and sequenced at Illumina Miseq platform with 2 \times 250 bp V2 Kits.

Sequencing: Line 187, please cite the 909R primer; also, inconsistencies in sequencing method used: Line 189 indicates MiSeq, however Lines 194 and 196 indicate that pyrosequencing was used. Please rectify; Lines 198-203: Open or closed-reference OTU picking? More details on bioinformatics please.

Response: We accept the suggestion and modified the manuscript accordingly.

Line 246: We cited the 909R primer.

Lines 257-264: MiSeq sequencing was used in our study. We rectified and rewrote the relevant content.

Line 272: Closed-reference OTU picking was used.

More details on bioinformatics were added (Lines 256-276).

Lines 256-276 : The DNA sequence data were processed using QIIME–Version 1.7.0 (<http://qiime.org/>) (Caporaso et al. 2010). Sequences were assigned to each sample based on the unique barcodes and the following quality-control criteria were implemented: sequences of length <200 bp were removed; a minimum average quality score of 30 was allowed; a maximum number of primer mismatches allowed was 2; no errors in barcode were allowed; no ambiguous bases were allowed (Zhang et al. 2016). Chimeras were removed using the Uchime algorithm (Edgar et al. 2011). Sequences classified as “unassigned” and “archaea” were also removed. Prior to analysis, samples were rarefied to the same sequence depth (7000 bacterial sequences per sample) using daisy-chopper.pl. Sequences were clustered into operational taxonomic units (OTUs) at the 97% sequence similarity threshold using a closed-reference OTU picking protocol (Edgar 2013, Rideout et al. 2014). Singletons were removed from the OTU table for downstream analysis. Finally, representative sequences from each OTU were selected for taxonomic assignment using the Ribosomal Database Project (RDP) 16S Classifier (Wang et al. 2007).

Results: Lines 253-259 and Figure S1, not sure why these are included? There is no analysis of the taxonomic data and it is not mentioned in the Discussion, therefore it seems superfluous.

Response: We agree and accepted the reviewer's comments and suggestions. Lines 253-259 were deleted. Figure S1 had been removed from the manuscript.

Figure 3: why is a polynomial being used to fit beta and gamma diversity? A unimodal relationship for alpha diversity is expected, however there's no clear justification for fitting the others using a polynomial function. Please support this decision.

Response: We thank the reviewer's good comments and suggestions.

The relations between (α -, β - and γ -) diversity and latitude were evaluated using linear and polynomial regressions. Akaike information criterion (AIC) was used to select and compare the goodness of fit of candidate linear and polynomial regression for the relation between (α - and β -) diversity and latitude. Adjusted Akaike information criterion AIC with correction for small sample size (AICc) was used to select the polynomial regression for γ diversity. By using AIC and AICc, we were able to select the final model that satisfies both simplicity and good statistical fit. We added this justification in revised manuscript, see lines 283-288.

Lines 347, 349 and 350: Latitude values changed for 45 and 48 to 47 and 46, respectively. Please be consistent.

Response: We accepted and corrected the inconsistent latitude values for throughout the manuscript. (Lines 441-448)

Fig 5: Please add a color key, it's not clear how colors should be interpreted.

Response: We thank the reviewer's suggestions. In the revised manuscript, we recalculated environmental heterogeneity according to the method provided by another reviewer.

Given that the updated method shows variations in total environmental heterogeneity (Fig. 7) and the coefficients of variation (CVs) of each environmental parameter (Supplementary Table S3) across all the locations, the original Figure 5 was deleted and the Figure 7 (Page 19) was created to show how overall environmental heterogeneity changed as well as CV of each environmental parameter.

Fig 6 legend: Bolivia and Missouri?!!

Response: Thank you for catching our oversight. This has been corrected in the modified manuscript. (Page 18, Fig. 6)

Line 505: I believe Figure S2 is the incorrect figure to reference here.

Response: Yes, it should be Fig.7 (Line 603). We corrected.

Please provide the environmental data in a separate table. It's impossible to interpret the author's statements about the homogenizing effect of pH on microbial communities at the 23N location (Lines 518-531) without seeing the pH data.

Response: We provided environmental data including soil pH in Supplemental information (Supplementary Table 3).

In our study, in 23° N location, homogenous selection dominated bacterial community

assembly under large variations in soil nitrogen (Fig.5b). Meanwhile, we also found the smallest species pool at 23° latitude location was associated with the lowest soil pH (Fig. 3c, Supplementary Table 3). One possible explanation is that variations of soil nutrients did not dominate community assembly processes under extreme soil pH conditions (Rousk et al. 2010, Tripathi et al. 2018). That is, the low soil pH at 23° N location narrows the niche of soil bacterial communities and consequently and consequently causes homogenizing effect.

Reference #46: Hubbell 2001: Please provide a chapter number.

Response: We have added a chapter number.

Hubbell SP. The unified neutral theory of biodiversity and biogeography (MPB-32). Princeton University Press, chapter 10, (2001).(Line 818)

Supplemental information:

- Table S1: The sample group information does not add significantly to the manuscript and I believe should be removed; why is there a footnote reading "Old growth with no sign of..." The footnote doesn't appear to be referenced in this Table. Please also include how far apart the plots are and how large the sites are.

Response: We agree and accepted the reviewer's comments and suggestions. We removed Table S1. Instead, we have provided spatial distribution maps of the plots for each location in the Supplementary Fig. 1. Detailed descriptions of the sampling sites and locations were also added in the Methods.

"Old growth with no sign of..." was deleted from the modified manuscript.

- Table S2: First row has a 48 in it? Is this a typo? Please also add horizontal and vertical lines between rows and columns as they are difficult to read; Same comment applies to panel b); Please also explain why some check-marks appear in bold font?

Response: We apologize for many errors in this table and we corrected all of them. The 48 in first row in Table S2 is a typo and has been removed. As suggested, we added horizontal and vertical lines between rows and columns in panels a and b. Marks in bold font have nothing meaning and was fixed.

- Fig S1: Fix typo.

Response: We deleted Fig S1, based on the reviewers' comment.

Reference

Bahram, M., F. Hildebrand, S. K. Forslund, J. L. Anderson, N. A. Soudzilovskaia, P. M. Bodegom, J. Bengtsson-Palme, S. Anslan, L. P. Coelho, and H. Harend. 2018. Structure and function of the global topsoil microbiome. *Nature* **560**:233.

Bahram, M., P. Kohout, S. Anslan, H. Harend, K. Abarenkov, and L. Tedersoo. Stochastic distribution of small soil eukaryotes resulting from high dispersal and drift in a local environment. *ISME Journal*.

Bell, T. 2010. Experimental tests of the bacterial distance–decay relationship. *The ISME journal* **4**:1357.

- Caporaso, J. G., J. Kuczynski, J. Stombaugh, K. Bittinger, F. D. Bushman, E. K. Costello, N. Fierer, A. G. Pena, J. K. Goodrich, and J. I. Gordon. 2010. QIIME allows analysis of high-throughput community sequencing data. *Nature methods* **7**:335.
- Catano, C. P., T. L. Dickson, and J. A. Myers. 2017. Dispersal and neutral sampling mediate contingent effects of disturbance on plant beta-diversity: A meta-analysis. *Ecology Letters* **20**:347-356.
- Chase, J. M., N. J. Kraft, K. G. Smith, M. Vellend, and B. D. Inouye. 2011. Using null models to disentangle variation in community dissimilarity from variation in α - diversity. *Ecosphere* **2**:1-11.
- Dexter, K. G., J. W. Terborgh, and C. W. Cunningham. 2012. Historical effects on beta diversity and community assembly in Amazonian trees. *Proceedings of the National Academy of Sciences* **109**:7787-7792.
- Edgar, R. C. 2013. UPARSE: highly accurate OTU sequences from microbial amplicon reads. *Nature methods* **10**:996.
- Edgar, R. C., B. J. Haas, J. C. Clemente, C. Quince, and R. Knight. 2011. UCHIME improves sensitivity and speed of chimera detection. *Bioinformatics* **27**:2194-2200.
- Ferrenberg, S., S. P. O'Neill, J. E. Knelman, B. Todd, S. Duggan, D. Bradley, T. Robinson, S. K. Schmidt, A. R. Townsend, and M. W. Williams. 2013. Changes in assembly processes in soil bacterial communities following a wildfire disturbance. *The ISME journal* **7**:1102-1111.
- Martiny, J. B., J. A. Eisen, K. Penn, S. D. Allison, and M. C. Horner-Devine. 2011. Drivers of bacterial β -diversity depend on spatial scale. *Proceedings of the National Academy of Sciences* **108**:7850-7854.
- Moran, P. A. 1950. Notes on continuous stochastic phenomena. *Biometrika* **37**:17-23.
- Morlon, H., G. Chuyong, R. Condit, S. Hubbell, D. Kenfack, D. Thomas, R. Valencia, and J. L. Green. 2008. A general framework for the distance-decay of similarity in ecological communities. *Ecology Letters* **11**:904-917.
- Myers, J. A., J. M. Chase, R. M. Crandall, and I. Jiménez. 2015. Disturbance alters beta - diversity but not the relative importance of community assembly mechanisms. *Journal of Ecology* **103**:1291-1299.
- Post, W. M., J. Pastor, P. J. Zinke, and A. G. Stangenberger. 1985. Global patterns of soil nitrogen storage. *Nature* **317**:613.
- Ranjard, L., S. Dequiedt, N. Chemidlin Prévost-Bouré, J. Thioulouse, N. P. A. Saby, M. Lelievre, P. A. Maron, F. E. R. Morin, A. Bispo, C. Jolivet, D. Arrouays, and P. Lemanceau. 2013. Turnover of soil bacterial diversity driven by wide-scale environmental heterogeneity. *Nature Communications* **4**:1434.
- Raup, D. M., and R. E. Crick. 1979. Measurement of faunal similarity in paleontology. *Journal of Paleontology*:1213-1227.
- Rideout, J. R., Y. He, J. A. Navas-Molina, W. A. Walters, L. K. Ursell, S. M. Gibbons, J. Chase, D. McDonald, A. Gonzalez, and A. Robbins-Pianka. 2014. Subsampled open-reference clustering creates consistent, comprehensive OTU definitions and scales to billions of sequences. *PeerJ* **2**:e545.
- Rousk, J., E. Bååth, P. C. Brookes, C. L. Lauber, C. Lozupone, J. G. Caporaso, R. Knight, and N. Fierer. 2010. Soil bacterial and fungal communities across a pH gradient in an arable soil. *The ISME journal* **4**:1340-1351.
- Stegen, J. C., X. Lin, J. K. Fredrickson, X. Chen, D. W. Kennedy, C. J. Murray, M. L. Rockhold, and A. Konopka. 2013. Quantifying community assembly processes and identifying features that

- impose them. *The ISME journal* **7**:2069-2079.
- Tamaki, H., C. L. Wright, X. Li, Q. Lin, C. Hwang, S. Wang, J. Thimmapuram, Y. Kamagata, and W.-T. Liu. 2011. Analysis of 16S rRNA amplicon sequencing options on the Roche/454 next-generation titanium sequencing platform. *PLoS One* **6**:e25263.
- Tripathi, B. M., J. C. Stegen, M. Kim, K. Dong, J. M. Adams, and Y. K. Lee. 2018. Soil pH mediates the balance between stochastic and deterministic assembly of bacteria. *The ISME journal* **12**:1072.
- Vannette, R. L., and T. Fukami. 2017. Dispersal enhances beta diversity in nectar microbes. *Ecology Letters* **20**:901-910.
- Wang, Q., G. M. Garrity, J. M. Tiedje, and J. R. Cole. 2007. Naive Bayesian classifier for rapid assignment of rRNA sequences into the new bacterial taxonomy. *Applied & Environmental Microbiology* **73**:5261.
- Wang, X.-B., X.-T. Lü, J. Yao, Z.-W. Wang, Y. Deng, W.-X. Cheng, J.-Z. Zhou, and X.-G. Han. 2017. Habitat-specific patterns and drivers of bacterial β -diversity in China's drylands. *The ISME journal* **11**:1345.
- Willig, M. R., D. M. Kaufman, and R. D. Stevens. 2003. Latitudinal gradients of biodiversity: pattern, process, scale, and synthesis. *Annual review of ecology, evolution, and systematics* **34**:273-309.
- Zhang, X., S. Liu, X. Li, J. Wang, Q. Ding, H. Wang, C. Tian, M. Yao, J. An, and Y. Huang. 2016. Changes of soil prokaryotic communities after clear-cutting in a karst forest: evidences for cutting-based disturbance promoting deterministic processes. *FEMS Microbiology Ecology* **92**:fiw026.

Reviewers' Comments:

Reviewer #1:

Remarks to the Author:

The authors did a great job with the revision of the MS. It has improved a lot in clarity and the previously detected problems have been mostly appropriately addressed. However, the increased clarity of the MS has shed light on some further problems.

First, following the advice of the other reviewer the authors simplified their terminology and instead of writing about, 'species pool', 'sizes of species pool' or 'neutral sampling effects' or 'sampling effects', they use only 'species pool' throughout the MS when referring to the regional species pool. However, the regional species pool has both a size, which is reflected in gamma-diversity, and a distribution (i.e., the different ratio of the different species within the regional species pool). In the case of microbial communities, the distribution of a species pool is essential as the abundance of species can vary in several orders of magnitude. Therefore, using the term 'species pool' for both can be misleading. Accordingly, I suggest the following. First, I think 'regional species pool' would be more adequate and conforming the field than simple 'species pool'. Second, the authors should always clarify whether they refer only to the richness of the regional species pool or also its distribution. I suggest that in the first case they always clearly specify that they refer to gamma-diversity, and only when they are considering the distribution of species too, refer to species pool. This use of terminology has to be clarified early on in the MS. Some examples where this simplified terminology leads to confusions: l. 25-27, l. 126-127, l. 548-551.

Although I couldn't check the R scripts used by the authors as they were not provided, they refer to the study by Kraft et al (2011) as the source and Kraft et al used log-normal distribution for the simulations. It has to be clear from this MS too whether the authors used log-normal distribution of species for the simulations of the regional species pool on beta-diversity, and the R codes should be provided. This is an important aspect. Again, I feel that the importance of species distribution has been omitted.

I highly appreciate that the authors created figure 1 to explain the conceptual background of their study. Conceptual figures are a great tool to increase clarity of studies especially in such a complex case. However, I think figure 1 is in several ways not the most appropriate. First, scenario 1 is partly inspired by the Supplementary figure 1 of Kraft et al. but in this MS the idea has been simplified. Connected to my previous problem, there is no information about the distribution of the species within the regional species pool (unlike in Kraft). The arrows going from the species pool to the local communities should reflect a random sampling, which depends on species richness and distribution. This all should be clear from the figure.

Another problem with the figure is that now the Scenario 2 looks like each type of assembly process happens only at the given gamma-diversity. I suggest therefore, to instead line these different processes horizontally making it clear that this are not gamma diversity dependent processes. Alternatively, this should go to a separate subfigure.

Furthermore, I have a problem with the terminology. The last assembly process in scenario 2 is 'stochastic process'. However, the assembly process expected in scenario 1 is also stochastic. This is a problem throughout the MS (e.g., l. 79-81) and needs to be clarified already in the hypothesis of the study. Namely, the primary hypothesis that this study is challenging is that beta-diversity depends on the regional species pool. In this case, beta-diversity is a result of random sampling of individuals from the regional species pool (where species have a defined distribution) into the local communities. Random sampling is per se a stochastic process. It is the same stochastic process that according to the figure is expected in the last panel of scenario 2. Stochastic processes consist of both regional (i.e., random dispersal of species from regional pool among localities) and local processes (i.e., drift related processes such as mortality or birth). This terminology should be clarified throughout the text.

A further problem with the figure and the entire MS (e.g., l. 139-141, l. 447-450) is that homogenizing dispersal is not considered as a potential assembly process. This process, which is

similar to mass effects in the metacommunity framework has to be added to the figure and considered throughout the MS when discussing lower than expected beta-diversity deviations. Finally, I think the right part of the figure could be improved. I still feel that showing the difference of 2 points as a next point along the x axis is misleading. I think simply calculating beta-deviation from the values on figure could be enough. Namely, explaining that beta-deviation = beta-observed - beta-expected, and then for each scenario writing beta-deviation < 0 or beta-deviation > 0.

Another major problem that I have with the MS is the evaluation of the beta-deviations. I couldn't identify whether these values were statistically evaluated. The difference between the expected and observed beta-diversities has to be tested and the p-values of the tests have to be presented. Otherwise these results are meaningless. For example, in the case of 23° the deviation could easily be non-significant.

The authors did consider spatial autocorrelation in the revised MS but then the spatially autocorrelated variables were not removed according to the supplementary tables. Why? This has to be justified. I also don't know (as the statistical analyses description is limited and no codes are provided) how the variance partitioning was performed. Why isn't there a joint fraction of spatial and environmental factors? That could help explain the importance of spatial autocorrelation.

Finally, I think there is a conceptual idea that the authors haven't addressed. They expect gamma-diversity to affect beta-diversity but I think this is more likely to happen the other way around: beta-diversity is influenced by local factors and that is that affects the regional gamma-diversity and species pool. For example, high environmental heterogeneity among localities leads to high beta-diversity, which might in return result in high gamma-diversity. I think this is a much more likely hypothesis in the case of soil microbiology. I know analyzing this hypothesis could require some major changes to the MS, which I am not proposing. However, I think that the opposite causation as a possibility should be mentioned in the introduction or discussion.

Detailed comments:

More details about the statistical analyses including R codes and versions have to be provided. For example, it is not clear how correction for species pool was performed (l. 25-27).

l. 72-74: Why would high dispersal increase overall species diversity? I would expect it to decrease.

l. 88-90: I don't really get this sentence...

l. 90-92: There are many studies about this. Maybe it is not clear yet but this sentence feels suggesting that it has been never studied.

l. 125: „coincide” feels a bit too high expectation, maybe „correlate” instead.

l. 184: these are quite precise values for being “approximately”.

l. 218: “1:5” specify what ratio this is. Volume?

Fig 2.: I suggest you mark the samples with their latitude value as use refer to them throughout the MS.

l. 244: change “was” to “were”.

l. 263: the applied sequencing depth is rather low for soil microbiomes. Please, provide rarefaction curves in the supplementary. Although, sampling of rare taxa is less important for beta-diversity

studies (See for example Szekely & Langenheder 2014 FEMS Microb ecol) but these has to be discussed.

I. 264: "daisychopper.pl". What is this? Is this an R function or qiime? Please, provide some context, citation and version numbers (not only here but for all tools used in the MS).

I. 273-276: were the sequences curated based on the annotations? Were non-microbial or non-classified sequences removed?

I. 295-297: There were difference in the case of beta-diversity whether the outlier was included or not. This has to be mentioned.

I. 330: "from" feels like it has to be deleted.

I. 391-392: Is 53° tropical?

I. 433: "from" feels unnecessary.

L 539: something is missing at the end of this sentence.

I. 547: change "kraft" to "Kraft".

I. 621: Delete one "and consequently"

I. 675-677: How does increased environmental heterogeneity decrease dispersal rates? Maybe it decreases the establishment success of dispersed species but by dispersal rate I would think of the mass of dispersed individuals. Maybe it is worth to be a bit more specific here.

Supplementary fig. 7: I think this correlation is led only by the outlier. Please present it without the outlier too and discuss it.

Reviewer #2:

Remarks to the Author:

I appreciate the authors' sincere efforts to address the reviewer comments, which have significantly improved the manuscript overall. The addition of primary metadata and maps showing the spatial distribution of sampling locations as well as the vastly improved details on methodology are particularly beneficial and were necessary. The thorough analyses and results are well-described and documented and the figures are improved. Overall the manuscript was much more enjoyable to read and interpret.

The major criticism of the manuscript that remains for me is the emphasis on the novelty of the research question and how it is being framed. Regarding the framing, the contrast of bacterial beta diversity patterns with macroorganisms seems disingenuous because the authors only refer to macroorganisms in the introduction and then return to it in the discussion. There are no specific hypotheses or expectations demonstrating how latitudinal diversity patterns are expected to differ between macro- and microorganisms. In short, the authors do not present a compelling argument for framing the manuscript in terms of differences between macro- and microorganisms.

Another weak part of the manuscript lies in Lines 82-97, where the authors outline 'empirical gaps', which read more like best practices for interpreting beta diversity patterns rather than true limitations to understanding. The authors seem to be arguing that many studies that use beta diversity are incorrectly interpreting their data because they don't properly account for gamma diversity, biogeography, or environmental filtering in their interpretations. These are more

concerns over scaling, since small-scale studies may justifiably ignore the influence of biogeography or gamma diversity. Furthermore, we do, in fact, know quite a bit about the spatially autocorrelated and biogeographic patterns of microbial diversity from numerous studies, as the authors also cite. The important point the authors should instead emphasize here is the application of these factors to a study of this geographic scale. Their argument would also be strengthened by referring to specific studies that the authors believe demonstrate a lack of considering these other factors on beta diversity patterns.

The novelty of this manuscript, to me, lies more in the geographic scale and depth of sampling and analysis rather than in the novel analytical approach or resulting conclusions.

Finally, while the writing is significantly improved, there are still some lingering spelling and grammatical errors that should be corrected.

Some examples:

Lines 122-142: Refer to the scenarios 1 and 2 in Figure 1 as appropriate.

Figure 1: Remove extra parentheses “)”

Line 377: The citation text probably needs to be removed

Line 433: remove ‘from’

Line 465: I think you need to change this to ‘bacteria’

Line 523: There is increasing evidence

Line 539: Soil bacterial what? Looks like sentence is incomplete.

Line 547: Capitalize “kraft”

Line 621: repeated “and consequently”

Dear Reviewers,

We provide detailed point-by-point responses to each of the reviewer's points in blue.

Thank you for your time.

Sincerely,

Shirong

Corresponding author, on behalf of all authors

Reviewer #1 (Remarks to the Author):

The authors did a great job with the revision of the MS. It has improved a lot in clarity and the previously detected problems have been mostly appropriately addressed. However, the increased clarity of the MS has shed light on some further problems.

First, following the advice of the other reviewer the authors simplified their terminology and instead of writing about, 'species pool', 'sizes of species pool' or 'neutral sampling effects' or 'sampling effects', they use only 'species pool' throughout the MS when referring to the regional species pool. However, the regional species pool has both a size, which is reflected in gamma-diversity, and a distribution (i.e., the different ratio of the different species within the regional species pool). In the case of microbial communities, the distribution of a species pool is essential as the abundance of species can vary in several orders of magnitude. Therefore, using the term 'species pool' for both can be misleading. Accordingly, I suggest the following. First, I think 'regional species pool' would be more adequate and conforming the field than simple 'species pool'. Second, I suggest that in the first case they always clearly specify that they refer to gamma-diversity, and only when they are considering the distribution of species too, refer to species pool. This use of terminology has to be clarified early on in the MS. Some examples where this simplified terminology leads to confusions: l. 25-27, l. 126-127, l. 548-551.

Response: We really appreciate the reviewer's good comments and suggestions, and agree that the regional species pool has both a size, which is reflected in γ -diversity and a species abundance distribution. In the revision, as suggested by the reviewer, 'regional species pool' was used only when considering both regional species richness and species distribution, instead of simply 'species pool'. γ -diversity refers to regional species richness, as a size of regional species pool. Regional species pool and γ -diversity were clarified in Lines 161-163. We also modified these terminologies accordingly throughout the manuscript. For example, see Lines 28, 29, 124, 159-160, 300-303, 322-331, 446-455, 552 and 559.

In this revision, we first estimated the effects of γ -diversity on β -diversity through

exploring the relationship between β -diversity and γ -diversity, using both a log-normal species abundance distribution and a uniform species abundance distribution based on the method suggested by Kraft et al (2011). Then, we examined how local community assembly processes influenced β -diversity after controlling for either one or both the two components of regional species pool by using two different null models (Tello et al. 2015, Vannette and Fukami 2017). The first null model only controls for γ -diversity (in this null model, the species pool is assumed as the observed number of species in a region; the regional species abundance distribution is randomized by re-assigning individuals to each species in the region with equal probability) and the second null model controls for both γ -diversity and species abundance distribution of regional species pool (the species pool is defined as the observed number of species and species abundance distribution in a region; the regional species abundance distribution is constrained to be the same in null and empirical datasets). The R code for the simulations and null models are provided as a supplementary file.

Although I couldn't check the R scripts used by the authors as they were not provided, they refer to the study by Kraft et al (2011) as the source and Kraft et al used log-normal distribution for the simulations. It has to be clear from this MS too whether the authors used log-normal distribution of species for the simulations of the regional species pool on beta-diversity, and the R codes should be provided. This is an important aspect. Again, I feel that the importance of species distribution has been omitted.

Response: In our analysis, we used a log-normal species abundance distribution for the simulations of the regional species pool (see R code). In addition, a uniform species abundance distribution was also used as suggested by Kraft et al (2011). Simulations show similar relationships for samples generated from log-normal and uniform abundance distributions (Supplementary Figure 8). We added the detailed description of species distribution in the revised version (see Lines 300-303, 438-444) and provided R codes for the simulations as a supplementary file.

I highly appreciate that the authors created figure 1 to explain the conceptual background of their study. Conceptual figures are a great tool to increase clarity of studies especially in such a complex case. However, I think figure 1 is in several ways not the most appropriate. First, scenario 1 is partly inspired by the Supplementary figure 1 of Kraft et al. but in this MS the idea has been simplified. Connected to my previous problem, there is no information about the distribution of the species within the regional species pool (unlike in Kraft). The arrows going from the species pool to

the local communities should reflect a random sampling, which depends on species richness and distribution. This all should be clear from the figure. Another problem with the figure is that now the Scenario 2 looks like each type of assembly process happens only at the given gamma-diversity. I suggest therefore, to instead line these different processes horizontally making it clear that this are not gamma diversity dependent processes. Alternatively, this should go to a separate subfigure.

Response: We would like to thank the reviewer's excellent comments and suggestions on conceiving the conceptual figure. We modified the figure accordingly. First, we added the information regarding the species abundance distribution of regional species pool. Second, the arrows going from regional species pool to local communities in Scenario I reflect a random sampling, which depends on regional species richness and abundance distribution. Third, different community assembly processes were lined horizontally in Scenario II as a separated subfigure. See Figure 1 in the revised manuscript.

Figure 1 Hypotheses tested in this study to illuminate the underlying drivers of bacterial β -diversity patterns along a gradient. Scenario I: If regional species pools dominate β -diversity pattern, then β -diversity will vary depending on γ -diversity and the distribution of species abundance. For example, with the increasing γ -diversity along the gradient, β -diversity increases accordingly. In this study, γ -diversity refers to regional species richness (a size of species pool), while regional species pool has both regional species richness and species abundance distribution.

Scenario II: If local community assembly processes dominate β -diversity pattern, the variations in β -diversity depend mainly on local community assembly processes. For instance, β -diversity is different due to different community assembly processes even when the observed γ -diversity is consistent. β -deviation is calculated as the difference between the observed and mean expected β -diversity, divided by the standard deviation (SD) of expected values. Throughout the figure, letters represent hypothetical species. Large ovals represent regional species pools and small circles with letters represent local communities that comprise a subset of species from the regional species pool. The arrows going from the species pools to the local communities in scenario I reflect random sampling. The arrows passing through each respective condition in scenario II represent heterogeneous selection, homogeneous selection, stochastic processes, homogenizing dispersal and dispersal limitation, respectively. Here, stochastic processes refer to the ecological processes that give rise to β -diversity patterns that are indistinguishable from random chance alone, such as ecological drift (Chase and Myers 2011). The shaded areas containing several hexagons represent environmental conditions, where hexagons with different colors represent environmental heterogeneity; a hexagon of a single color represents homogeneity. Dash lines perpendicular to the direction of arrows in the shaded area represent the dispersal limitation; dash lines parallel to the direction of arrows represent homogenizing dispersal.

Furthermore, I have a problem with the terminology. The last assembly process in scenario 2 is 'stochastic process'. However, the assembly process expected in scenario 1 is also stochastic. This is a problem throughout the MS (e.g., l. 79-81) and needs to be clarified already in the hypothesis of the study. Namely, the primary hypothesis that this study is challenging is that beta-diversity depends on the regional species pool. In this case, beta-diversity is a result of random sampling of individuals from the regional species pool (where species have a defined distribution) into the local communities. Random sampling is per se a stochastic process. It is the same stochastic process that according to the figure is expected in the last panel of scenario 2. Stochastic processes consist of both regional (i.e., random dispersal of species from regional pool among localities) and local processes (i.e., drift related processes such as mortality or birth). This terminology should be clarified throughout the text.

Response: We agree with the reviewer's comments that 'stochastic process' in Scenario I is inappropriate. In Scenario I, β -diversity is a result of random sampling of individuals from the regional species pool into the local communities. Thus, the arrows should reflect a 'random sampling'. Scenario II shows how different community assembly processes influence β -diversity. Here, 'stochastic processes' is used to refer to the ecological processes that give rise to β -diversity pattern that are indistinguishable from random chance alone, such as random dispersal of species from regional pool among localities and drift processes. We clarified this terminology

in Figure 1 and throughout the text. See Lines 75, 177-179, 305 and 348-349.

A further problem with the figure and the entire MS (e.g., l. 139-141, l. 447-450) is that homogenizing dispersal is not considered as a potential assembly process. This process, which is similar to mass effects in the metacommunity framework has to be added to the figure and considered throughout the MS when discussing lower than expected beta-diversity deviations.

Response: We agree and accept the reviewer's suggestion and homogenizing dispersal was added as a potential assembly process (see Figure 1). Also, this process was considered when discussing the lower than the expected β -diversity (see Line 138-141, 352-354).

Finally, I think the right part of the figure could be improved. I still feel that showing the difference of 2 points as a next point along the x axis is misleading. I think simply calculating beta-deviation from the values on figure could be enough. Namely, explaining that $\text{beta-deviation} = \text{beta-observed} - \text{beta-expected}$, and then for each scenario writing $\text{beta-deviation} < 0$ or $\text{beta-deviation} > 0$.

Response: We accept the reviewer's good suggestion on the improvement of the conceptual Figure. We explained $\beta\text{-deviation} = (\text{observed } \beta - \text{expected } \beta) / \text{SD expected } \beta$, and then used $\beta\text{-deviation} < 0$ or $\beta\text{-deviation} > 0$ for each scenario in the revised Figure 1.

Another major problem that I have with the MS is the evaluation of the beta-deviations. I couldn't identify whether these values were statistically evaluated. The difference between the expected and observed beta-diversities has to be tested and the p-values of the tests have to be presented. Otherwise these results are meaningless. For example, in the case of 23° the deviation could easily be non-significant.

Response: We thank the reviewer's thoughtful comments on the evaluation of the β -deviations. In the revised manuscript, we statistically evaluated the difference between the observed β -diversity and the distribution of 999 expected β -diversities from null models for each sample (see R code) and *P*-values of the tests was presented in Source data. The proportions of significant observed β -diversity compared to the expected β -diversity (including significantly lower than expected β -diversity, significantly greater than expected β -diversity and no different from expected β -diversity) in each region were shown in Supplementary Table 4.

Also, one-sample *t*-test was used to compare mean β -deviation against zero for 60 samples in each region and the *P*-values of the tests were presented in Supplementary

Table 5. The statistical results showed the mean β -deviations were significantly different from zero in all the regions after controlling for variations in either γ -diversity or in regional species pool (Supplementary Table 5).

Supplementary Table 4 The proportions of significant observed β -diversity compared to the expected β -diversity in each region ($P < 0.05$) (a) after controlling for γ -diversity and (b) after controlling for regional species pool.

(a)

Regions	18 °N	23 °N	29 °N	31 °N	33 °N	36 °N	39 °N	42 °N	45 °N	47 °N	53 °N
Significantly lower than expected β -diversity	28.3%	46.7%	0	0	23.3%	51.7%	58.3%	56.7%	51.7%	20%	6.7%
Significantly greater than expected β -diversity	60%	30%	98.3%	96.7%	75%	45%	36.7%	36.7%	41.7%	76.7%	86.7%
No significantly different from expected β -diversity	11.7%	23.3%	1.7%	3.3%	1.7%	3.3%	5%	6.6%	6.6%	3.3%	6.6%

(b)

Regions	18 °N	23 °N	29 °N	31 °N	33 °N	36 °N	39 °N	42 °N	45 °N	47 °N	53 °N
Significantly lower than expected β -diversity	3.3%	0	0	0	0	0	1.7%	0	0	0	0
Significantly greater than expected β -diversity	96.7%	100%	100%	100%	100%	100%	96.6%	100%	100%	100%	100%
No significantly different from expected β -diversity	0	0	0	0	0	0	1.7%	0	0	0	0

Supplementary Table 5 One-sample t -test was used to compare mean β -deviation against zero for 60 samples in each region (a) after controlling for γ -diversity and (b) after controlling for regional species pool.

(a)

Regions	18 °N	23 °N	29 °N	31 °N	33 °N	36 °N	39 °N	42 °N	45 °N	47 °N	53 °N
Mean β -deviation	26.85	-6.07	46.57	48.2	36.19	-15.32	-20.56	-19.72	-19.79	31.09	48.4
P values	< 0.001	0.043	< 0.001	< 0.001	< 0.001	0.018	< 0.001	< 0.001	0.003	< 0.001	< 0.001

(b)

Regions	18 °N	23 °N	29 °N	31 °N	33 °N	36 °N	39 °N	42 °N	45 °N	47 °N	53 °N
Mean β -deviation	124.92	62.45	128.3	148.6	133.53	70.83	63.31	74.06	73.12	115.57	162.09
P values	< 0.001	< 0.001	< 0.001	< 0.001	< 0.001	< 0.001	< 0.001	< 0.001	< 0.001	< 0.001	< 0.001

The authors did consider spatial autocorrelation in the revised MS but then the spatially autocorrelated variables were not removed according to the supplementary tables. Why? This has to be justified. I also don't know (as the statistical analyses description is limited and no codes are provided) how the variance partitioning was

performed. Why isn't there a joint fraction of spatial and environmental factors? That could help explain the importance of spatial autocorrelation.

Response: We thank the reviewer's good suggestions on spatial autocorrelation. In the revised manuscript, the spatially autocorrelated variables were removed. After excluding spatially autocorrelated variables, the effects of environmental factors on community assembly and β -diversity were recalculated, and the results were shown in the revised supplementary Table 2, supplementary Table 3, Figure 6 and Figure 7.

There was a joint fraction of spatial and environmental factors. The joint fraction of spatial and environmental factors was added in the revised Figure 6. The variance partitioning was performed using 'varpart' function in R vegan package. We added the statistical analyses in the revised manuscript (see Lines 376-379), and provided R code for variance partitioning analysis and forward-selection stepwise regression analysis as a supplementary file.

Finally, I think there is a conceptual idea that the authors haven't addressed. They expect gamma-diversity to affect beta-diversity but I think this is more likely to happen the other way around: beta-diversity is influenced by local factors and that is that affects the regional gamma-diversity and species pool. For example, high environmental heterogeneity among localities leads to high beta-diversity, which might in return result in high gamma-diversity. I think this is a much more likely hypothesis in the case of soil microbiology. I know analyzing this hypothesis could require some major changes to the MS, which I am not proposing. However, I think that the opposite causation as a possibility should be mentioned in the introduction or discussion.

Response: We agree with the reviewer's comment on the possible effects of β -diversity on the regional species pool. We added this hypothesis in the discussion. See lines 573-576.

Detailed comments:

More details about the statistical analyses including R codes and versions have to be provided. For example, it is not clear how correction for species pool was performed (l. 25-27).

Response: More details regarding the statistical analyses are provided in the revised manuscript. See Lines 300-303, 322-354 and 366-379. In addition, to clearly show how the correction for species pool was performed, R codes including the two null models and statistical analyses were provided as a supplementary file.

l. 72-74: Why would high dispersal increase overall species diversity? I would expect

it to decrease.

Response: Thank you for catching our editorial oversight. We agree that high dispersal decrease β -diversity and overall species diversity, and thus this expression has been accordingly revised. See Lines 68-70.

l. 88-90: I don't really get this sentence...

Response: This sentence is not clear and we rephrased these sentences for improved clarity in the revised manuscript. See Lines 77-87.

l. 90-92: There are many studies about this. Maybe it is not clear yet but this sentence feels suggesting that it has been never studied.

Response: We are sorry about this ambiguous expression and we revised these sentences. See Lines 77-87.

l. 125: „coincide” feels a bit too high expectation, maybe „correlate” instead.

Response: We changed and ‘coincide’ was replaced by ‘correlate’ as suggested. See Lines 123.

l. 184: these are quite precise values for being “approximately”.

Response: We deleted “approximately”. See Line 190.

l. 218: “1:5” specify what ratio this is. Volume?

Response: It was 1:5 (w/v, Weight : Volume), and we added this in the revised manuscript. See Line 213.

Fig 2.: I suggest you mark the samples with their latitude value as use refer to them throughout the MS.

Response: We have made the corresponding changes as suggested throughout the MS, see Figure 2.

l. 244: change “was” to “were”.

Response: We accepted and revised as suggested. See Line 241.

l. 263: the applied sequencing depth is rather low for soil microbiomes. Please, provide rarefaction curves in the supplementary. Although, sampling of rare taxa is less important for beta-diversity studies (See for example Szekely & Langenheder 2014 FEMS Microb ecol) but these has to be discussed.

Response: We thank the reviewer for this helpful comment on the applied sequencing

depth. We provided rarefaction curves (see Supplementary Figure 2). Although the rarefaction curves were not saturated for observed species, the distinctions among different regions along the latitude were observed. Also, β -diversity can be sufficiently described and explained by common taxa than rare taxa (Székely and Langenheder 2014). The variations in β -diversity among different regions could be compared based on this sequencing depth. This information was added in the revised manuscript. Line 266-270.

Supplementary Figure 2 Rarefaction curves for bacterial diversity. Lines represent observed species (a) and Shannon diversity (b). Error bars around each point indicate the standard deviation among 60 samples.

I. 264: “daisy-chopper.pl”. What is this? Is this an R function or qiime? Please, provide some context, citation and version numbers (not only here but for all tools used in the MS).

Response: We added some description, “daisy-chopper.pl” is a Perl script. Samples were rarefied to the same sequence depth using the Perl script daisy-chopper.pl (Gilbert et al. 2009, McMurdie and Holmes 2014). We also added the context and the citation for this tool as well as other tools used in the MS. See Lines 261-266.

I. 273-276: were the sequences curated based on the annotations? Were non-microbial or non-classified sequences removed?

Response: Yes, the sequences were curated based on the annotations. Sequences classified as “unassigned” and “non-bacterial” were removed.

I. 295-297: There were difference in the case of beta-diversity whether the outlier was included or not. This has to be mentioned.

Response: The manuscript has been revised to account for the difference in

β -diversity due to the outlier. We examined the distribution of β -diversity within each region using raw data. Although some potential outliers of β -diversity were found (see Supplementary Figure 3), there was no difference in the β -diversity pattern along the latitude whether the outlier was included or not (see Figure 3 and Supplementary Figure 6). Therefore, our analyses with the complete dataset were presented to ensure the high statistical power achieved by the robust experimental design. See Lines 280-299.

I. 330: “from” feels like it has to be deleted.

Response: We rephrased this sentence. See Line 340.

I. 391-392: Is 53 °tropical?

Response: We revised and clarified that tropical is at 18 ° and cold temperate zone is at 53 °N, respectively. See Line 395.

I. 433: “from” feels unnecessary.

Response: We deleted it. See Line 438.

L 539: something is missing at the end of this sentence.

Response: We revised and added the corresponding content to complete this sentence. See Lines 532-533.

I. 547: change “kraft” to “Kraft”.

Response: We changed it. See Line 555.

I. 621: Delete one “and consequently”

Response: We deleted it. See Line 603.

I. 675-677: How does increased environmental heterogeneity decrease dispersal rates? Maybe it decreases the establishment success of dispersed species but by dispersal rate I would think of the mass of dispersed individuals. Maybe it is worth to be a bit more specific here.

Response: Thanks for this thoughtful suggestion. We added some more specific discussion in the revised manuscript. Greater environmental heterogeneity could decrease the establishment success of dispersed species and then reduce the similarity of bacterial community composition, leading to a higher β -diversity. (See 613-615).

Supplementary fig. 7: I think this correlation is led only by the outlier. Please present

it without the outlier too and discuss it.

Response: Yes, the reviewer is right. We provided the figure to show the correlation without the outlier, and there was no significant relationship between γ -diversity and soil pH along the latitude when excluding the outlier (see Response Figure 1). That is, the significant correlation between γ -diversity and soil pH along the latitude is led only by the outlier, indicating the correlations between them is inconclusive based on available data in this study. However, given that our study focused on how regional species pool and local community assembly processes influence bacterial β -diversity pattern, we deleted the analysis between γ -diversity and soil pH at the latitudinal scale.

Response Figure 1 The relationship between γ -diversity and soil pH along the latitude excluding the outlier (23 °N region).

Reviewer #2 (Remarks to the Author):

I appreciate the authors' sincere efforts to address the reviewer comments, which have significantly improved the manuscript overall. The addition of primary metadata and maps showing the spatial distribution of sampling locations as well as the vastly improved details on methodology are particularly beneficial and were necessary. The thorough analyses and results are well-described and documented and the figures are improved. Overall the manuscript was much more enjoyable to read and interpret.

The major criticism of the manuscript that remains for me is the emphasis on the novelty of the research question and how it is being framed. Regarding the framing, the contrast of bacterial beta diversity patterns with macroorganisms seems disingenuous because the authors only refer to macroorganisms in the introduction and then return to it in the discussion. There are no specific hypotheses or expectations demonstrating how latitudinal diversity patterns are expected to differ between macro- and microorganisms. In short, the authors do not present a

compelling argument for framing the manuscript in terms of differences between macro- and microorganisms.

Response: We greatly appreciate the reviewer's critical comments on the novelty of the research question. We are sorry for our inappropriate representation causing misunderstanding. After careful perusal of the literatures regarding bacterial diversity, we recognize that previous studies have done extensive work in framing research question and analytical approach for addressing patterns of bacterial β -diversity along environmental gradients and geographical gradients. We agree that the contrasting bacterial β -diversity patterns with macro-organisms was not adequately addressed and investigated in our study.

In the revised manuscript, we rephrased the research question and reframed our hypothesis. In this regard, we conceived a hypothesis to address soil bacterial β -diversity pattern along the latitudinal gradient in NSTEC, focusing on the characteristics of bacterial communities and their influencing factors, rather than simply comparing microorganisms with macroorganisms. Accordingly, we also addressed the hypothesis and discussed the possible reasons in the discussion. See Lines 114-118 and 533-542.

Lines 114-118 in the Introduction: 1. Is there a latitudinal gradient in β -diversity of soil bacteria in eastern China? Earlier studies indicate that the turnover of bacterial communities (β -diversity) is highly correlated with the variations in soil variables (Ranjard et al. 2013, Wang et al. 2017). Considering a large variability and complexity of soil factors along the latitude, we hypothesize that the observed β -diversity pattern will not exhibit a latitudinal gradient.

Lines 533-542 in the Discussion:

As expected, β -diversity of soil bacteria did not exhibit a latitudinal gradient (Fig.3b, Supplementary Fig.7b). The reasons for the lack of an apparent latitudinal gradient in β -diversity are manifold, among which is that diversity and abundance of bacterial communities are constrained primarily by edaphic variables (Fierer and Jackson 2006, Delgado - Baquerizo et al. 2016, Wang et al. 2017). Our results also showed that variations in β -diversity along the latitudinal gradient were significantly correlated with the heterogeneity of soil parameters (Fig. 7b), which is consistent with other studies showing that the turnover of soil bacterial composition is driven by changes in soil conditions (Ranjard et al. 2013, Dini-Andreote et al. 2015). These findings indicate that latitude, as a complex gradient along which many environmental

variables are changing, is not sufficient for inferring large scale controls on geographical patterns of bacterial diversity.

Another weak part of the manuscript lies in Lines 82-97, where the authors outline 'empirical gaps', which read more like best practices for interpreting beta diversity patterns rather than true limitations to understanding. The authors seem to be arguing that many studies that use beta diversity are incorrectly interpreting their data because they don't properly account for gamma diversity, biogeography, or environmental filtering in their interpretations. These are more concerns over scaling, since small-scale studies may justifiably ignore the influence of biogeography or gamma diversity. Furthermore, we do, in fact, know quite a bit about the spatially autocorrelated and biogeographic patterns of microbial diversity from numerous studies, as the authors also cite. The important point the authors should instead emphasize here is the application of these factors to a study of this geographic scale. Their argument would also be strengthened by referring to specific studies that the authors believe demonstrate a lack of considering these other factors on beta diversity patterns.

Response: We thank the reviewer for the suggestions on critical comments on how to demonstrate a lack of addressing the β -diversity patterns. We agree that small-scale studies may justifiably ignore the influence of biogeography or γ -diversity, and thus, have modified the expression in the revised manuscript (see Lines 83-85). We also recognize that the spatially autocorrelated and biogeographic patterns of microbial diversity have been greatly explored by the numerous previous studies and therefore, we rephrased these sentences and demonstrated the application of some known factors plus unknown factors to this study at geographical scale by referring to specific studies, while highlighting a better understanding of the mechanism underlying the β -diversity patterns along the latitudinal gradient. See Lines 77-87.

Lines 77-87 in the revised manuscript:

Recent studies on soil bacterial β -diversity patterns showed that the similarity of bacterial communities decline with increasing geographic distance and environmental heterogeneity (Martiny et al. 2011, Ranjard et al. 2013, Wang et al. 2017), and that variations in bacterial β -diversity are strongly correlated with variations in environmental factors including soil pH, soil C and N contents (Dini-Andreote et al. 2015, Wang et al. 2017, Tripathi et al. 2018). As environmental variables often vary along latitudinal gradients (Delgado - Baquerizo et al. 2016), this could be the main cause of variations in local community assembly and further influence the relative

importance of deterministic and stochastic processes across different regions. Alternatively, variations in the regional species pool itself could also result in variations in β -diversity wherein γ -diversity and species distribution significantly vary among different regions (Kraft et al. 2011, Myers et al. 2015). Thus, there is a clear need to distinguish the relative importance of local community assembly processes from regional species pool, and this will help to ascertain the mechanism underlying the geographical pattern of soil bacterial community.

The novelty of this manuscript, to me, lies more in the geographic scale and depth of sampling and analysis rather than in the novel analytical approach or resulting conclusions.

Response: We fully agreed with the reviewer's critical comments. To better clarify the novel features of this study, we modified the Introduction, reframed our hypothesis, and accordingly addressed the latitudinal gradient in the discussion. See Lines 48-53, 77-87, 114-118 and 522-542.

Finally, while the writing is significantly improved, there are still some lingering spelling and grammatical errors that should be corrected.

Some examples:

Lines 122-142: Refer to the scenarios 1 and 2 in Figure 1 as appropriate.

Response: We carefully checked and modified these sentences refer to the scenarios 1 and 2 in Figure 1. See lines 119-141.

Figure 1: Remove extra parentheses “)”

Response: We accepted and removed it. See line 158.

Line 377: The citation text probably needs to be removed

Response: We accepted and removed it. See Line 385-386.

Line 433: remove 'from'

Response: We removed it. See Line 438.

Line 465: I think you need to change this to 'bacteria'

Response: We modified as suggested. See Line 473.

Line 523: There is increasing evidence

Response: We modified as suggested. See Line 518.

Line 539: Soil bacterial what? Looks like sentence is incomplete.

Response: We added the corresponding content. See Lines 532-533.

Line 547: Capitalize “kraft”

Response: We revised as suggested, see Line 555.

Line 621: repeated “and consequently”

Response: We revised as suggested, see Line 603.

Reference

- Chase, J. M., and J. A. Myers. 2011. Disentangling the importance of ecological niches from stochastic processes across scales. *Philosophical Transactions of the Royal Society of London B: Biological Sciences* **366**:2351-2363.
- Delgado - Baquerizo, M., F. T. Maestre, P. B. Reich, P. Trivedi, Y. Osanai, Y. R. Liu, K. Hamonts, T. C. Jeffries, and B. K. Singh. 2016. Carbon content and climate variability drive global soil bacterial diversity patterns. *Ecological monographs* **86**:373-390.
- Dini-Andreote, F., J. C. Stegen, J. D. van Elsas, and J. F. Salles. 2015. Disentangling mechanisms that mediate the balance between stochastic and deterministic processes in microbial succession. *Proceedings of the National Academy of Sciences*:201414261.
- Fierer, N., and R. B. Jackson. 2006. The diversity and biogeography of soil bacterial communities. *Proceedings of the National Academy of Sciences* **103**:626-631.
- Gilbert, J. A., D. Field, P. Swift, L. Newbold, A. Oliver, T. Smyth, P. J. Somerfield, S. Huse, and I. Joint. 2009. The seasonal structure of microbial communities in the Western English Channel. *Environmental microbiology* **11**:3132-3139.
- Kraft, N. J., L. S. Comita, J. M. Chase, N. J. Sanders, N. G. Swenson, T. O. Crist, J. C. Stegen, M. Vellend, B. Boyle, and M. J. Anderson. 2011. Disentangling the drivers of β diversity along latitudinal and elevational gradients. *Science* **333**:1755-1758.
- Martiny, J. B., J. A. Eisen, K. Penn, S. D. Allison, and M. C. Horner-Devine. 2011. Drivers of bacterial β -diversity depend on spatial scale. *Proceedings of the National Academy of Sciences* **108**:7850-7854.
- McMurdie, P. J., and S. Holmes. 2014. Waste not, want not: why rarefying microbiome data is inadmissible. *PLoS computational biology* **10**:e1003531.
- Myers, J. A., J. M. Chase, R. M. Crandall, and I. Jiménez. 2015. Disturbance alters beta - diversity but not the relative importance of community assembly mechanisms. *Journal of Ecology* **103**:1291-1299.
- Ranjard, L., S. Dequiedt, N. Chemidlin Prévost-Bouré J. Thioulouse, N. P. A. Saby, M. Lelievre, P. A. Maron, F. E. R. Morin, A. Bispo, C. Jolivet, D. Arrouays, and P. Lemanceau. 2013. Turnover of soil bacterial diversity driven by wide-scale environmental heterogeneity. *Nature Communications* **4**:1434.
- Székely, A. J., and S. Langenheder. 2014. The importance of species sorting differs between habitat generalists and specialists in bacterial communities. *FEMS Microbiology Ecology*

87:102-112.

- Tello, J. S., J. Myers, M. J. Macia, A. F. Fuentes, L. Cayola, G. Arellano, M. I. Loza, V. Torrez, M. Cornejo, and T. Miranda. 2015. Elevational Gradients in β -Diversity Reflect Variation in the Strength of Local Community Assembly Mechanisms across Spatial Scales. *PLoS One* **10**:0121458.
- Tripathi, B. M., J. C. Stegen, M. Kim, K. Dong, J. M. Adams, and Y. K. Lee. 2018. Soil pH mediates the balance between stochastic and deterministic assembly of bacteria. *The ISME journal* **12**:1072.
- Vannette, R. L., and T. Fukami. 2017. Dispersal enhances beta diversity in nectar microbes. *Ecology Letters* **20**:901-910.
- Wang, X.-B., X.-T. Lü, J. Yao, Z.-W. Wang, Y. Deng, W.-X. Cheng, J.-Z. Zhou, and X.-G. Han. 2017. Habitat-specific patterns and drivers of bacterial β -diversity in China's drylands. *The ISME journal* **11**:1345.

Reviewers' Comments:

Reviewer #1:

Remarks to the Author:

The authors did an amazing job with the revision of the MS. I have no major comments only minor ones.

p. 3, l. 78: Use „declines“ instead of „decline“.

p. 3, l. 80-83: This sentence is too long and difficult to follow. I suggest breaking it up into 2 sentences.

p. 6., l. 163: I suggest replacing “has” with something like “reflects” or “depicts” or similar.

p. 11., l. 286: Remove “any”.

p. 11., l. 296-297: Although in respect of the final conclusions you drove from the results there was no difference between the analyses with and without outliers, it would be appropriate to mention here that there was some difference in the curvilinear model of beta-diversity with (Fig. 3b) and without the outlier 23° region (Fig. 7b).

p. 13., l. 350-354: I suggest instead of writing “zero indicates heterogeneous selection or dispersal limitation is dominated in shaping beta-diversity”, to write “zero indicates the dominance of heterogeneous selection or dispersal limitation in shaping beta-diversity”. The same for the second half of the sentence.

Fig. 7b: Consider changing the y-axis to not start at 0 as for Fig. 7a and c, as this could make the differences described in l. 392-393 visually more obvious.

Fig. 5a: The light gray expected values are almost impossible to see. I suggest using the same dark gray as on Fig. 5c.

p. 23., l. 635-638: A very likely further explanation could be that in the low environmental heterogeneity/low explained variation regions, other unmeasured environmental factors were important both in defining environmental heterogeneity and beta-diversity. Although you mention unmeasured environmental variables in l. 628, here you only mention ecological processes. Please, make this part a bit clearer.